# Intervening Anchor Token: Decoding Strategy in Alleviating Hallucinations for MLLMs

**Feilong Tang**[1,2*], **Zile Huang**[1,2*], **Chengzhi Liu**[4], **Qiang Sun**[2,3], **Harry Yang**[1,2], **Ser-Nam Lim**[2,5]
[1]HKUST, [2]Everlyn AI, [3]University of Toronto, [4]University of Liverpool, [5]UCF
`tangfeilong1@gmail.com, sernam@ucf.edu`

## Abstract

Multimodal large language models (MLLMs) offer a powerful mechanism for interpreting visual information. However, they often suffer from hallucinations, which impede the real-world usage of these models. Existing methods attempt to alleviate this issue by designing special decoding strategies that penalize the summary tokens. However, these methods lack analysis of the relationship between hallucination and the summarization mechanism of LLMs. Interestingly, we find that penalizing summary tokens is not necessary: merely intervening in the query-key parameters variance, without costing extra inference time, still alleviates hallucinations. Specifically, we explore the causes of hallucinations by analyzing localized self-attention patterns called *"anchor" tokens* and define the attention localization degree of the model as *token propagation probabilities*. Our analysis reveals that over-propagation of anchor tokens occurs when the distribution of eigenvalues of the query and key matrices has a non-zero mean and a polarized variance, leading to excessive dependence on anchor tokens while neglecting vision information and describing the image content with hallucination. Based on the observation, we propose a versatile plug-and-play decoding strategy, *Dynamic Token Propagation Mechanism* (**TAME**), to alleviate excessive propagation by dynamically intervening in the eigenspectrum variance of the attention weight, thereby alleviating hallucinations without relying on complex decoding strategies. Extensive experiments reveal a correlation between the eigenspectrum and hallucinations across various MLLMs and show that TAME reduces the percentage of hallucinated objects. Code released at https://github.com/Everlyn-Labs/ANTRP.

## 1 Introduction

Recent advancements in multi-modal large language models (MLLMs) (Xue et al., 2025; Zhang et al., 2023a; Liu et al., 2024d; Bai et al., 2023; Dai et al., 2023a; Liu et al., 2024c;b; Dong et al., 2024) have propelled general-purpose foundation models to unprecedented capabilities. These advancements have equipped MLLMs with the ability to process images as inputs, enabling highly dynamic and contextually rich interactions. The advanced functionality of MLLMs allows them to be adept at a variety of vision-related tasks(Black et al., 2023; Zhang et al., 2023b; Li et al., 2024a), while seamlessly handling more complex tasks such as content comprehension (Lai et al., 2024) and generation (Geng et al., 2024). Despite their remarkable versatility, MLLMs often suffer from hallucinations. Specifically, these models tend to generate fabricated or incorrect outputs in response to user-provided images and prompts, often producing irrelevant or nonsensical information, or misidentifying objects in terms of colors, quantities, or locations that do not exist in the image.

Various approaches (Wang et al., 2024a; Yin et al., 2023; Zhou et al., 2023) have been proposed to mitigate hallucinations in MLLMs. These methods often incur substantial additional costs, including the annotation budget for extra instruction data for training (Liu et al., 2023a), or the integration of external knowledge or models. Conversely, other approaches focus on decoding strategy optimization to penalize the knowledge aggregation patterns, avoiding training but doubling or even tripling inference time. OPERA (Huang et al., 2024) introduces a penalty-based re-decoding approach to alleviate the over-trust summary token issue. Contrastive Decoding (CD) strategies adjust logits for

---

*Equal Contribution. Work done during internship at Everlyn AI.

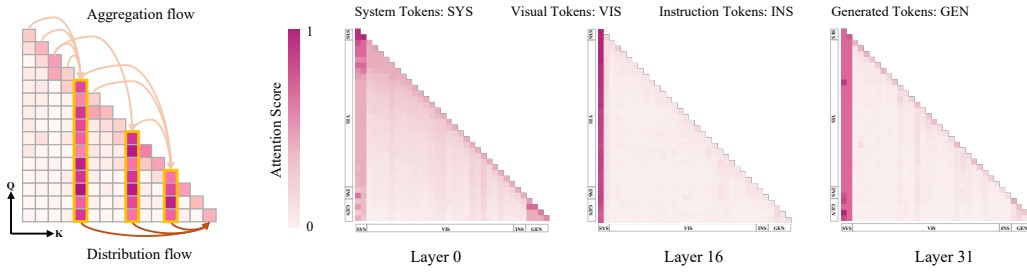

(a) Token Propagation Pattern    (b) The Attention Maps Across Different Layers During the Decoding Process of LLaVA1.5-7B.

Figure 1: (a): Illustration of the token propagation pattern in the Self-attention Query-Key Matrix, where anchor tokens (highlighted in orange boxes) aggregate and distribute knowledge. (b): The attention maps across layers during the decoding process of LLaVA1.5-7B show that in initial layers, attention is evenly distributed. However, in deeper layers, attention aggregates mainly towards system, instruction, and generated tokens, while attention to image tokens becomes sparse. Detailed attention allocation is provided in Appendix A. *These findings indicate that anchor tokens, potentially caused by the QK-parameters, attract most of the attention and contribute to hallucinations.*

next-token prediction using contrastive techniques. Vision CD applies Gaussian noise (Leng et al., 2024) or ablates visual inputs (Favero et al., 2024) to amplify language priors, while Instruction CD (Wang et al., 2024b; Jiang et al., 2024) introduces noise by adding random words, contradictory commands, or truncated instructions. Despite their effectiveness, these methods incur computational burdens and impede the deployment of MLLMs on personal devices. Furthermore, the relationship between hallucination and the inherent summarization mechanism of LLMs remains unexplored.

Recent studies (Wang et al., 2023; Pang et al., 2024) have shown that token information flow aggregates to a few "anchor tokens", from which the model extracts information to make predictions, facilitating tokens interaction patterns and in-context learning, as illustrated in Figure 1 (a). However, (Huang et al., 2024; Wu et al., 2024) empirically found that hallucinations stem from an over-reliance on partial anchor tokens. Specifically, the limited anchor tokens cannot retain the rich visual information provided by the entire context. During the transmission of information between anchor tokens, the visual information becomes attenuated as the length of the generated text increases. Their findings suggest that subsequent tokens neglect the initial visual input leading to hallucinations caused by the model bias. Up until now, discussions about anchor tokens have been conducted independently, each with slightly different interpretations. As a result, our understanding of the blessing and curse of anchor tokens remains elusive.

To delve deeper into this phenomenon, we analyze the attention maps of the first, middle, and final layers during the decoding process of a model response as illustrated in Figure 1 (b). As attention can be regarded as a token mixer, in the shallow layers, attention scores are more uniformly distributed across different tokens. Whereas in the deeper layers, system prompts display vertical strong lines that take up most of the attention scores (which we call *localized* attention). Our statistical analysis reveals a highly imbalanced attention distribution: in the deep layers, attention is focused on these anchor tokens, leading to significantly reduced attention on the image tokens themselves. This results in the model generating content inconsistent with the actual facts in images. Based on these observations, we propose the following two key research questions: (Q1) *When* are tokens localized or uniform? (Q2) *How* does anchor tokens affect the generation of hallucinations?

In this paper, we characterize self-attention token patterns through the attention weight matrix to investigate the root causes of hallucinations. First, we define the concept of anchor tokens through *token propagation probability* (Section 2), which describes the likelihood of a specific input token propagating its information to other tokens within the information flow of LLM. Our rigorous statistical analysis reveals that hallucinatory captions tend to exhibit higher token propagation probabilities. Then, we demonstrate that the propagation pattern of anchor tokens can be characterized by the eigenspectrum of the attention weight matrix (Section 3 and 4) by (Bao et al., 2024). Specifically, proper-propagation of anchor tokens enhances expressivity when the query-key eigenspectrum has a non-zero mean and a small variance. However, over-propagation triggers hallucinations when the variance becomes polarized. To alleviate this issue, we propose a versatile plug-and-play decoding strategy, *Dynamic Token Propagation Mechanism* (**TAME**), which reduces the over-propagation of anchor tokens through dynamically intervening in the eigenspectrum variance (Section 5). Interest-

ingly, we find that penalizing summary tokens is not necessary: merely intervening in the query-key parameters variance, without incurring extra inference time, still alleviates hallucinations. Lastly, with extensive experiments, we observe a correlation between the eigenspectrum and hallucinations in various MLLMs, and demonstrate that TAME reduces the percentage of hallucinated objects.

## 2 TOKEN PROPAGATION PROBABILITY

This section scrutinizes the root causes of hallucinations in vision-language models through comprehensive statistical analyses of token propagation probability and hallucination. We also provide a rigorous theoretical explanation that complements our empirical findings on hallucinations.

**Notations.** MLLMs generate text in a auto-regressive manner by progressively predicting the probability distribution of the next token. In this section, we represent the input as $X = \{x_1, x_2, \ldots, x_T\} \in \mathbb{R}^{h \times T}$, where $x_i \in \mathbb{R}^h$ is the embedding of the $i$-th token and $T$ is the number of tokens. The correct answer is denoted as $y$, and the model-generated sequence, consisting of $N$ tokens, is represented as $Z = \{z_1, z_2, \ldots, z_N\} \in \mathbb{R}^{h \times N}$. Specifically, the probability of generating the $i$-th token $z_i$ is modeled as $p(z_i|s_{<i}, X)$, where $1 \leq i \leq N$ and $s_{<i}$ represent the sequence of previously generated tokens before the $i$-th token. Several decoding strategies are developed based on $p$, including Greedy Decoding and Beam Search. The decoded token is concatenated to the end of the original input text for the next round of generation, continuing until the process concludes.

During autoregressive generation, the model employs a self-attention mechanism to capture dependencies between tokens. At the $\ell$-th layer, for each attention head, the self-attention is defined as:

$$A^\ell = S\left(\frac{(X^{\ell-1})^\top W_{QK} X^{\ell-1}}{\sqrt{d}}\right), \quad U^\ell = W_V X^{\ell-1} A^\ell, \tag{1}$$

where $W_V \in \mathbb{R}^{h \times h}$ represents the value weight matrix, $W_{QK} = W_Q W_K^\top \in \mathbb{R}^{T \times T}$ is the combined query-key weight matrix, and $\sqrt{d} > 0$ is a temperature scaling factor. $S$ denotes the softmax function. At the $\ell$-th layer, $U^\ell$ represents the updated token embeddings after applying the value matrix $W_V$ and attention scores to the input embeddings in $X^{\ell-1}$.

**Uniform vs. Localized Softmax.** We employ Sparsemax (Martins & Astudillo, 2016), a piecewise linear alternative to Softmax, to streamline the computation of Gaussian moments while preserving the attention structure of the original Softmax. To linearize $S(\kappa)_i$ where $\kappa \in \mathbb{R}^h$ is a input vector, we perform a Taylor expansion at the origin, yielding:

$$\epsilon^i = \nabla_i S(0) = \frac{1}{T} e^i - \frac{1}{T^2} 1, \quad \epsilon_0^i = S(0)_i = \frac{1}{T}, \tag{2}$$

where $\epsilon^i$ and $\epsilon_0^i$ represent the expansion coefficients. We then approximate $S$ using a piecewise linear function $\widetilde{S}$, as follows:

$$S(\kappa)_i \approx \max\{0, \min\{1, \langle \epsilon^i, \kappa \rangle + \epsilon_0^i\}\} = \langle \widetilde{\epsilon}^i, \kappa \rangle + \widetilde{\epsilon}_0^i = \widetilde{S}(\kappa)_i. \tag{3}$$

This indicates that the $i$-th input token is activated when $\delta_i = \langle \epsilon^i, \kappa \rangle + \epsilon_0^i \in [0, 1]$. Otherwise, $\widetilde{S}(\kappa)_i = \widetilde{\epsilon}_0^i$, effectively preventing the input token $x_i$ from contributing to the self-attention mechanism. Building on this, we quantify the likelihood of activation for the $i$-th token in the Softmax function to estimate the extent to which its information propagates to other tokens.

**Definition 1** (Token Propagation Probability). *Suppose that $W_{QK}$ is independent of X. For each $i \in [T]$, the token propagation probability of the $i$-th token is defined as:*

$$\rho_i = \mathbb{P}\{\delta_i \in [0, 1]\}, \tag{4}$$

*where $\kappa = X^\top W_{QK} x_T / \sqrt{d}$, and the randomness originates solely from the input tokens X.*

When only a few $\rho_i$ are significantly greater than zero, the softmax function behaves as localized, meaning the self-attention mechanism (Eq. 1) is dominated by a few anchor tokens. Conversely, uniform softmax, which produces similar $\rho_i$ values, results in equal contributions from most tokens.

Based on the definition of the Token Propagation Probability $\rho_i$, we compare the distributions of $\rho_i$ between hallucinatory and non-hallucinatory captions (see Appendix C.3 for details). As shown in Figure 2, hallucinatory captions tend to exhibit higher token propagation probabilities, which suggests a stronger association between object hallucination and higher propagated tokens.

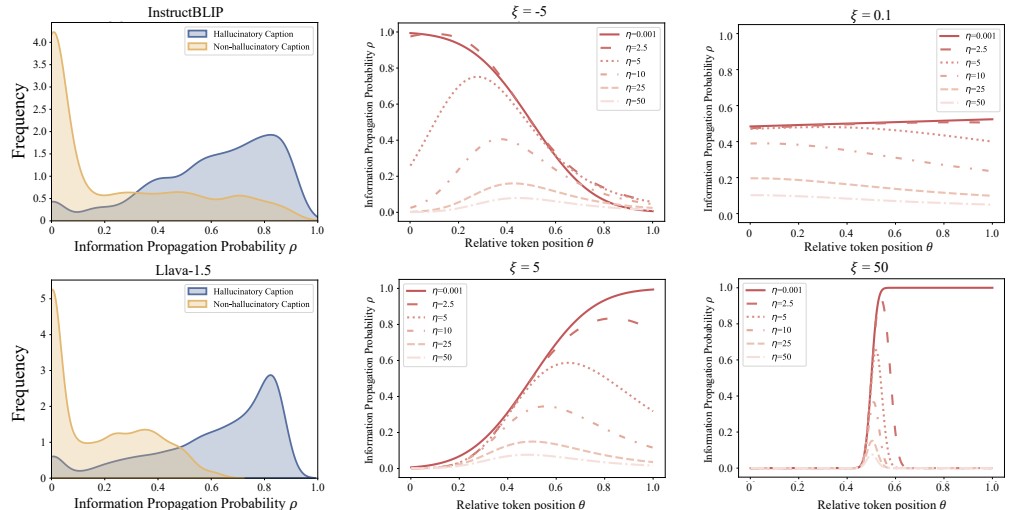

Figure 2: **(Left)**: Comparison of token propagation probability between hallucinatory and non-hallucinatory captions generated by LLaVA-1.5 and InstructBLIP models. **(Right)**: The plots of the Token Propagation Probability $\rho(\theta)$ for varying $\phi$ and $\omega$. The horizontal axis indicates relative token position $\theta = i/T$ ( $i$ : token index, $T$ : number of tokens) by (Bao et al., 2024).

## 3   WHEN ARE TOKENS LOCALIZED OR UNIFORM?

**Assumption 1** (Gaussian Token Distribution). *Assume that the tokens $(\mathbf{x}_t)_{t \geq 1}$ are independent and identically distributed (i.i.d.) random vectors drawn from a multivariate Gaussian distribution:*

$$\mathbf{x}_t \sim \mathcal{N}(\mu, \Sigma), \quad \text{for all } t \geq 1, \tag{5}$$

*where $\mu$ is the mean vector, and $\Sigma$ is the covariance matrix.*

To derive $\rho_i$, note that $\delta_i$ is a linear combination of multiple random variables. By the Central Limit Theorem, it can be approximated as a normal distribution with mean and variance:

$$\mu^i = \mathbb{E}\left[\delta_i\right]; \quad v^i = \mathrm{Var}\left(\delta_i\right). \quad \delta_i \sim \mathcal{N}(\mu^i, v^i). \tag{6}$$

**Proposition 1.** *Suppose that $W_{QK}$ is symmetric and independent from $X$. Under Assumption 1, for $i \in [T]$, the mean $\mu^i$ and variance $v^i$ with the input $\kappa = X^\top W_{QK} \mathbf{x}_T / \sqrt{d}$ as:*

$$\mu^i = c_1 \frac{\mathrm{tr}(W)}{\sqrt{d}} + o(1); \quad v^i = c_2 \frac{\mathrm{tr}\left(W^2\right)}{d} + o(1), \tag{7}$$

*where $W$ denotes the weighted covariance matrix as $W = W_{QK}\Sigma$. $\mathrm{tr}(W)$ represents the trace of $W$. $c_1$ and $c_2$ are constants, with $c_1 = \frac{i}{T} - \frac{1}{2}$ and $c_2 = \frac{2i^2}{T^2} + \frac{7}{12}$.*

When $W_{QK}$ is asymmetric, we redefine the token propagation probability using the symmetrized matrix $\left(W_{QK} + W_{QK}^\top\right) / 2$ in the following proposition. The $\mathrm{tr}(W)$ equals the sum of its eigenvalues $\mathrm{tr}(W) = \sum_{i=1}^h w_i$, where $w_1, w_2, \ldots, w_h$ are the eigenvalues of $W$.

**Proposition 2.** *Since $\mu^i$ and $v^i$ depend on the relative token location $i/T$, we extend $i/T$ continuously to $\theta \in [0, 1]$, and thus extend token propagation probability $\rho_i$ to $\rho : [0, 1] \to [0, 1]$ as:*

$$\rho(\theta) = \frac{1}{2}\mathcal{E}\left(\frac{\left(\theta - \frac{1}{2}\right)\phi}{\zeta(\theta)}\right) - \frac{1}{2}\mathcal{E}\left(\frac{\left(\theta - \frac{1}{2}\right)\phi - \frac{1}{\omega}}{\zeta(\theta)}\right), \tag{8}$$

*where $\phi = \mathrm{tr}(W)/\sqrt{\mathrm{tr}\left(W^2\right)}$, $\omega = \sqrt{\mathrm{tr}\left(W^2\right)}/\sqrt{d}$, with ranges $\phi \in [-\sqrt{d}, \sqrt{d}]$ and $\omega \in (0, \infty)$. $\zeta(\theta) = \sqrt{2\left(2\theta^2 + \frac{7}{12}\right)}$. $\mathcal{E}$ denotes the error function.*

**Remark:** When $W$ is independent of $X$, $\phi$ and $\omega$ can be considered independent variables, as the eigenspectrum scale of $\mathrm{tr}\left(W^2\right)$ can be adjusted within the bound (8) once the eigenspectrum of $W$

is given. $\phi$ measures the variance spread across eigenvalues, while $\omega$ represents the eigenvalue scale relative to the matrix dimension, indicating correlation strength between dimensions.

Figure 2 illustrates $\rho(\theta)$ with different $\phi$ and $\omega$, leading to several key observations: *(i) Localization.* $\rho(\theta)$ concentrates on fewer tokens as $\omega$ increases. By contrast, $\rho(\theta)$ behaves relatively uniformly regardless of $\omega$ for small $|\phi|$. *(ii) Location focus.* For small $\omega$, as $\phi$ increases, $\rho(\theta)$ assigns weight to late-site tokens. Conversely, for negative $\phi$, it focuses on early-site tokens. When $\omega$ increases, $\rho(\theta)$ localizes around the middle of the sequence for sufficiently large $\phi$. *(iii) Vanishing propagation.* As $\omega$ increases, $\rho(\theta)$ diminishes to zero for any $\theta \in [0, 1]$ regardless of $\phi$.

**Proposition 3.** *$\rho(\theta)$ satisfies the following properties by (Bao et al., 2024).*
*1. (Tokens Localized) When $\phi\omega \to r$ significantly deviates from zero, such that $|r| \gg 2$, the signal propagation probability $\rho(\theta)$ will concentrate at specific positions in the sequence.*
*2. (Tokens Uniform) With $\omega$ held as a finite value, as $|\phi|$ approaches zero, $\rho(\theta)$ approaches a constant value for any $\theta \in [0, 1]$.*
*3. (Vanishing Propagation) With $\phi$ fixed as a finite value, as $\omega$ increases indefinitely, $\rho(\theta)$ diminishes to zero for all $\theta \in [0, 1]$.*

**Remark.** Proposition 3 indicates that the behavior of the token propagation probability $\rho(\theta)$ is closely tied to the interaction between $\omega$ and $\phi$. As $\phi\omega = \mathrm{tr}(\mathrm{W})/\sqrt{d}$, we focus on the eigenspectrum of W, where the eigenvalues $(w_i)_{i \in [h]}$ are considered as samples from a distribution with mean $\mathrm{tr}(\mathrm{W}) = \sum_{i=1}^{d} w_i$ and scale $\mathrm{tr}(\mathrm{W}^2) = \sum_{i=1}^{h} w_i^2$ (assuming W is real diagonalizable).

Firstly, the condition $\omega \to 0$ indicates that the scale $\mathrm{tr}(\mathrm{W}^2)$ approaches zero (see Proposition. 2). Secondly, since $\phi\omega = \mathrm{tr}(\mathrm{W})/\sqrt{d} \to r \gg 2$, it follows that $\mathrm{tr}(\mathrm{W}) \gg 2\sqrt{d}$, meaning $\mathrm{tr}(\mathrm{W})$ is significantly different from zero. Combining these insights, we conclude that $\rho$ localizes when the eigenspectrum concentrates around a non-zero mean. This localization is more likely when the embedding dimension $d$ is large, allowing the eigenvalue sum $\mathrm{tr}(\mathrm{W})$ to remain significantly non-zero while the scale $\mathrm{tr}(\mathrm{W}^2)$ stays close to zero (*i.e.,* each eigenvalue is close to zero). Therefore, increasing the embedding dimension $d$ facilitates attention localization. Conversely, according to Proposition 3's assertion of uniformity, as $\phi$ approaches zero, $\rho(\theta)$ varies less across different token positions $\theta$. In this limit, when $\mathrm{tr}(\mathrm{W}) \to 0$, $\rho$ becomes uniform across positions.

---

**A1: When are tokens localized or uniform?**

• Localization: $\rho$ becomes localized when $\mathrm{tr}\left(\mathrm{W}^2\right)$ is close to zero while $|\mathrm{tr}(\mathrm{W})|$ is significantly different from zero; *i.e.,* the eigenspectrum of W concentrates around a non-zero mean.

• Uniform: $\rho$ becomes uniform when $\mathrm{tr}(\mathrm{W})$ is close to zero while $\mathrm{tr}\left(\mathrm{W}^2\right)$ remains finite, *i.e.,* the eigenspectrum of W has zero mean with finite variance.

• Vanishing: $\rho$ uniformly tends to zero when $\mathrm{tr}\left(\mathrm{W}^2\right)$ is sufficiently large; *i.e.,* the eigenspectrum of W has an infinitely large variance.

---

## 4 How does anchor token affect MLLMs?

**Proper-Propagation of Anchor Tokens Enhances Expressivity**: In self-attention blocks, *rank collapse* (Dong et al., 2021) indicates that the output matrix $U^\ell$ in Eq. 1 converges to a rank-1 matrix as $L \to \infty$, *i.e.,* $\lim_{L \to \infty} A^\ell = \mathbf{z}\mathbf{1}^\top$, where $\mathbf{z}$ is a non-zero vector and $\mathbf{1}^\top$ is an all-ones matrix. In this scenario, the attention matrix becomes uniform, causing the attention distributions of all input tokens to converge to the same value. This prevents the model from distinguishing between different input information, resulting in a gradual loss of diversity and expressiveness. (Dong et al., 2021) linked uniformity to the spectral properties of the weight matrix $W$, demonstrating that when the $\ell_1$ norm $\|\mathrm{W}_{\mathrm{QK}}\|_1$ of the matrix is large, the convergence to a rank-1 matrix slows down. This implies that when attention is appropriately propagated across a few anchor tokens, the localized attention distribution can guide the model to more effectively capture subtle feature differences, thereby leading to better expressivity. The connection between $\|\mathrm{W}_{\mathrm{QK}}\|_1$ and $|\mathrm{tr}(\mathrm{W})|$ is given as:

$$\frac{|\mathrm{tr}(\mathrm{W})|}{\sqrt{d}\|\Sigma\|_2} \leq \|\mathrm{W}_{\mathrm{QK}}\|_2 \leq \|\mathrm{W}_{\mathrm{QK}}\|_{\mathrm{F}} \leq \|\mathrm{W}_{\mathrm{QK}}\|_1, \tag{9}$$

where the first inequality is due to the bound (8) and the Cauchy-Schwarz inequality, it is sufficient to increase $|\mathrm{tr}(\mathrm{W})|$ under fixed $\mathrm{tr}\left(\mathrm{W}^2\right)$ to enhance expressivity.

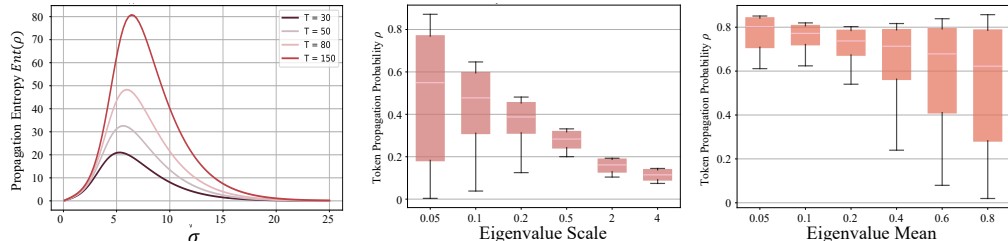

Figure 3: **(Left):** Propagation Entropy under different token length $T$. $\sigma = \|W_{QK}\|_2 \|XX^\top\|_2$. **(Right):** The box plots of the averaged Token Propagation Probability $\rho$ with different value of Eigenvalue Scale and Eigenvalue Mean. The experiments for Eigenvalue Scale are conducted with a fixed Eigenvalue Mean, and vice versa. With stronger Scale or smaller Mean, the $\rho$ increases.

**Over-Propagation of Anchor Tokens Triggers Hallucinations**: In self-attention blocks, the average *Shannon entropy* of the columns of the attention matrix $A^\ell$ (see Eq. 1) decreases, *i.e.,* $\lim_{L \to \infty} H_{\text{avg}}(A^\ell) \to 0$ where $H_{\text{avg}}(\cdot)$ measures the uniformity of the attention distribution. Intuitively, low attention entropy leads to localized attention. This concept is similar to ours. (Zhai et al., 2023; Bao et al., 2024) propose that low attention entropy leads to instability in transformer training and inference. They advocate for avoiding overly concentrated attention and demonstrated that the lower bound of attention entropy is a unimodal function of $\|W_{QK}\|_2$. In MLLMs, when token entropy approaches zero, the attention distribution becomes overly concentrated, meaning the model excessively relies on a few anchor tokens. This results in generated outputs depending more on the summarized information of these anchor tokens, rather than on the integrated information from visual and linguistic tokens in the context as in Figure 2. This demonstrates that over-propagation of anchor tokens leads to hallucinations. Therefore, we next investigate the properties of propagation entropy $Ent(\rho)$. We show in the next theorem that $Ent(\rho)$ is directly connected to $\|W_{QK}\|_2$.

**Theorem 1** (Propagation Entropy). *Let* $\sigma = \|W_K W_Q^\top\|_2 \|XX^\top\|_2$, *and* $\beta = \exp\left(-\sigma\sqrt{\frac{T}{T-1}}\right)$. *The propagation entropy* $Ent(\rho)$ *holds that:*

$$Ent(\rho) = \sigma \log(1 + (T-1)\beta) + \frac{\sigma^2 \sqrt{T(T-1)}\beta}{1 + (T-1)\beta}, \tag{10}$$

*where* $Ent(\rho)$ *represents that lower entropy increases the likelihood of over-propagation of anchor tokens, following a unimodal pattern in* $\sigma$, *and vanishing as* $\|W_{QK}\|_2 \to 0$ *or* $\infty$ *as illustrated in Figure 3(Left). Propagation entropy increases with* $\|W_{QK}\|_2$ *up to a peak, then decreases, being lowest at extreme values of* $\|W_{QK}\|_2$. *If* $|\operatorname{tr}(W)|$ *is moderate, propagation entropy stays near the peak. To mitigate over-propagation of anchor tokens, it is sufficient to control* $\operatorname{tr}(W^2)$ *under a fixed* $\operatorname{tr}(W)$, *constrain the eigenspectrum using the inequality:* $\|\Sigma^{-1}\|_F \sqrt{\operatorname{tr}(W^2)} \geq \|W_{QK}\|_2$.

**Remark:** We aim to observe the correlation between the eigenspectrum and the hallucinations triggered by the over-propagation of anchor tokens. As illustrated in Figure 3 (Right), $\rho$ localizes with smaller scales and larger means, which is consistent with the conclusion above. As the eigenvalue scale increases, the mean of $\rho$ decreases, with smaller scales yielding wider, more variable distributions, while larger scales lead to more stable, smaller values. Conversely, smaller means result in more concentrated $\rho$ distributions, whereas larger means cause greater dispersion and variability in token propagation. The results indicate that intervening in the scale and mean of W-eigenspectrum could be an effective way to mitigate over-propagation or propagation vanishing.

---

**A2: How Does Anchor Token Affect the MLLM Models?**

• *Enhance Expressivity*: Maximizing $|\operatorname{tr}(W)|$ while keeping $\operatorname{tr}(W^2)$ fixed allows anchor tokens to maintain complexity and capture subtle feature differences.

• *Trigger Hallucinations*: Polarization of $\operatorname{tr}(W^2)$ with fixed $|\operatorname{tr}(W)|$ reduces propagation entropy, causing over-propagation of anchor tokens and triggering hallucinations.

---

## 5 INTERVENING ANCHOR TOKEN PROPAGATION

The previous analysis demonstrates that anchor tokens significantly impact the expressive capability and hallucination phenomena of MLLMs through the eigenspectrum properties of the attention weight matrix. Specifically, moderate propagation of anchor tokens enhances the model's expressivity, while over-propagation leads to hallucinations. These findings highlight that controlling the eigenspectrum of the query-key weight matrix (W) can effectively regulate the propagation intensity of anchor tokens. Compared to intervening in $W = W_{QK}\Sigma$, we choose to directly adjust the eigenspectrum of $W_{QK}$. This is because W introduces the covariance matrix $\Sigma$, which remains unchanged during inference, making it sufficient to adjust $W_{QK}$ to flexibly control attention propagation. Hence, merely intervening in the $W_{QK}$ helps the model avoid triggers for hallucinations.

To achieve this goal, we propose a versatile plug-and-play decoding strategy, *Dynamic Token Propagation Mechanism* (TAME), to alleviate excessive propagation by dynamically intervening the eigenspectrum variance of the attention weight, thereby reducing hallucinations without relying on complex decoding strategies. Hence, the $W_{QK}$ is reparameterized as:

$$\widehat{W}_{QK} = (1 + \frac{\gamma}{log(\eta + \xi)})W_{QK}, \tag{11}$$

where $\eta$ denotes $\mathrm{tr}(W_{QK}^2)$, and $\gamma$ controls the eigenspectrum scale of $W_{QK}$. $\xi$ is a small constant like $10^{-6}$. The logarithm prevents excessive scaling when $\eta$ is near 0, decreasing the total scaling when $\eta$ is large and increasing it when $\eta$ is small, thus adjusting the variance of $W_{QK}$. Our method degrades to original weight when $\gamma$ is set to 0. The algorithm is provided in Appendix E.

## 6 EXPERIMENTS

### 6.1 SETUP

**Models.** For our evaluation, we choose four of the most exemplary MLLMs: InstructBLIP (Dai et al., 2023b), MiniGPT-4 (Zhu et al., 2023), LLaVA-1.5 (Liu et al., 2024b), and Shikra (Chen et al., 2023a). These MLLM models can be broadly categorized into two groups. The first group, comprising InstructBLIP and MiniGPT-4, utilizes the Q-former (Li et al., 2023b) to effectively bridge the vision and text modalities by representing images with only 32 tokens. In contrast, the second group, which includes LLaVA-1.5 and Shikra, employs linear projection layers to align the features of both modalities, requiring a larger number of image tokens 256 or even 576 as input for the MLLMs. Additionally, all these MLLM models incorporate a robustly pretrained vision encoder, such as CLIP (Radford et al., 2021) or EVA (Fang et al., 2023), alongside a pretrained language model like LLaMA (Touvron et al., 2023a) or Vicuna (Chiang et al., 2023).

**Baselines and Benchmark.** Since decoding strategies in a training-free manner, we compare seven decoding methods: Sampling (Top-p=1) Decoding, Greedy Decoding, Visual Contrastive Decoding (VCD) (Leng et al., 2024), Instruction Contrastive Decoding (ICD) (Wang et al., 2024b), Beam Search Decoding (Sutskever, 2014), beam-search-based OPERA (Huang et al., 2024), and SID (Huo et al., 2024). The proposed TAME can be seamlessly integrated into different decoding strategies, it simply replaces the variance of the self-attention $W_{QK}$ parameters of LLM. We evaluate TAME's performance of mitigating hallucinations on both long descriptions and simplified VQA answers. *i.e.,* Caption Hallucination Assessment with Image Relevance (CHAIR) (Rohrbach et al., 2018) and Polling-based Object Probing Evaluation (POPE) (Li et al., 2023c)).

### 6.2 QUANTITATIVE RESULTS

**CHAIR Evaluation on Hallucinations.** The CHAIR metric is a specialized evaluation tool developed to assess the issue of object hallucination in image captioning tasks. Specifically, CHAIR measures the extent of object hallucination in a given image description by calculating the proportion of objects mentioned in the description that are not present in the corresponding ground-truth label set. The evaluation consists of two distinct dimensions: $\mathrm{CHAIR}_S$, which operates at the sentence level, and $\mathrm{CHAIR}_I$, which operates at the image level. These metrics are represented as $C_S$ and $C_I$, and their detailed formulations are provided in Appendix C.2. We perform the CHAIR evaluation on the MSCOCO dataset (Lin et al., 2014), which includes over 300,000 images and annotations for 80 object categories. Following the Baseline method, we randomly select 500 images from the validation set of COCO 2014 and prompt variousMLLMs) with the query "`Please describe this`

Table 1: CHAIR hallucination evaluation results on four MLLM models. Denote CHAIRS as $C_S$ and CHAIRI as $C_I$. Smaller values indicate less hallucinations. TAME as a plug-and-play method.

| | Max New Tokens: 512 | | | | | | | | Max New Tokens: 64 | | | | | | | |
| | LLaVA-1.5 | | InstructBLIP | | Shikra | | MiniGPT4 | | LLaVA-1.5 | | InstructBLIP | | Shikra | | MiniGPT4 | |
| Methods | $C_S$ | $C_I$ | $C_S$ | $C_I$ | $C_S$ | $C_I$ | $C_S$ | $C_I$ | $C_S$ | $C_I$ | $C_S$ | $C_I$ | $C_S$ | $C_I$ | $C_S$ | $C_I$ |
|---|---|---|---|---|---|---|---|---|---|---|---|---|---|---|---|---|
| Sampling | 51.3 | 16.8 | 55.6 | 24.2 | 48.9 | 14.7 | 33.6 | 19.4 | 21.4 | 7.9 | 31.2 | 14.2 | 28.0 | 10.5 | 20.0 | 8.7 |
| **+TAME** | **47.7** | **15.9** | **53.1** | **21.7** | **45.4** | **12.8** | **30.7** | **17.9** | **19.5** | **7.1** | **28.7** | **12.2** | **25.3** | **9.5** | **18.8** | **6.7** |
| Greedy | 49.6 | 14.4 | 57.2 | 15.8 | 47.1 | 13.9 | 35.7 | 25.5 | 22.6 | 7.2 | 30.0 | 14.5 | 22.0 | 7.0 | 24.2 | 8.2 |
| **+TAME** | **47.3** | **14.1** | **54.7** | **14.8** | **42.0** | **11.7** | **33.2** | **18.4** | **20.1** | **6.8** | **28.9** | **13.6** | **19.2** | **6.9** | **22.1** | **6.9** |
| Beam | 48.0 | 14.3 | 54.3 | 16.1 | 46.6 | 12.5 | 32.1 | 17.8 | 23.4 | 8.2 | 31.6 | 13.8 | 20.2 | 6.4 | 18.8 | 5.9 |
| **+TAME** | **45.2** | **14.0** | **52.0** | **14.7** | **43.1** | **11.2** | **29.4** | **15.8** | **21.2** | **7.3** | **29.2** | **14.0** | **17.7** | **6.0** | **17.9** | **5.6** |
| VCD | 51.7 | 15.6 | 51.0 | 16.7 | 48.0 | 14.0 | 30.4 | 14.2 | 23.6 | 8.6 | 30.0 | 11.2 | 27.0 | 10.4 | 22.0 | 10.6 |
| **+TAME** | **43.8** | **14.1** | **48.0** | **14.5** | **44.6** | **13.4** | **29.1** | **13.5** | **21.5** | **7.3** | **28.8** | **9.9** | **24.8** | **9.9** | **18.3** | **9.5** |
| ICD | 47.4 | 13.9 | 46.3 | 15.3 | 47.3 | 14.1 | 32.6 | 13.1 | 21.0 | 8.7 | 32.2 | 10.6 | 27.5 | 11.8 | 20.0 | 8.7 |
| **+TAME** | **44.5** | **13.6** | **42.8** | **13.5** | **45.9** | **12.2** | **28.6** | **12.7** | **19.4** | **7.6** | **29.4** | **7.6** | **25.3** | **10.2** | **19.2** | **7.8** |
| OPERA | 46.4 | 13.0 | 47.1 | 12.4 | 46.4 | 12.7 | 26.4 | 10.7 | 21.8 | 8.2 | 28.4 | 9.7 | 22.6 | 12.8 | 22.6 | 8.2 |
| **+TAME** | **38.2** | **12.0** | **41.7** | **10.0** | **42.6** | **10.5** | **25.3** | **9.7** | **18.8** | **6.9** | **19.7** | **7.6** | **18.5** | **8.4** | **19.7** | **7.6** |
| SID | 44.2 | 12.2 | 42.3 | 12.4 | 44.8 | 12.8 | 28.5 | 11.7 | 20.7 | 8.4 | 26.0 | 8.6 | 29.8 | 11.7 | 23.1 | 10.7 |
| **+TAME** | **32.2** | **9.6** | **36.5** | **9.2** | **35.8** | **10.0** | **22.4** | **8.6** | **17.5** | **7.3** | **21.9** | **8.4** | **23.6** | **10.7** | **18.5** | **8.1** |

Figure 4: GPT-4 assisted hallucination evaluation (Zhao et al., 2023) results on VG-100K dataset analyze six key metrics: sentences per image (SPI), words per image (WPI), hallucinated sentences per image (HSPI), hallucinated words per image (HWPI), hallucinated sentences ratio (HSR), and hallucinated words ratio (HWR). Note that larger SPI and WPI, smaller HSPI, HWPI, HSR and HWR are better. Larger radar indicates better performance.

image in detail" to generate descriptions. To ensure a fair evaluation, we impose two different maximum token limits, as the length of generated sequences can significantly affect CHAIR scores ($C_S$ and $C_I$) (Li et al., 2023c). As shown in Table 1 our TAME obviously surpasses all of baselines decoding methods in both $C_S$ and $C_I$ metrics. Especially on LLaVA-1.5, our method achieves approximately a 27.1% improvement on SID. TAME consistently performs well in both long and short description generation.

**GPT-4 Assisted Evaluation.** While CHAIR is a robust metric for assessing object-existence-level hallucinations, it is limited in detecting other forms of hallucination, such as those involving object attributes, locations, or relationships. HalluBench (Zhao et al., 2023) represents a more advanced benchmark, utilizing detailed object-level descriptions from the VG dataset (Krishna et al., 2017) as ground-truth, and relying on GPT-4 to evaluate hallucinations in generated descriptions. In this process, the detailed object-level descriptions are compiled into a comprehensive but unordered summary of the image, and GPT-4 is carefully prompted to assess hallucinations in the descriptions generated by MLLMs, sentence by sentence. MLLMs are prompted with "Please describe this image in detail," with the maximum token limit set to 512.

As illustrated in Figure 4, our TAME method demonstrates a significant reduction in the occurrence of hallucinated sentences and words when generating descriptions for each image. Specifically, TAME achieves a 34.3% improvement over greedy decoding in terms of the hallucinated sentence ratio (HSR), and a 12.7% reduction in the number of hallucinated words per image (HWPI) compared to OPERA. This suggests that TAME, as a plug-in, effectively mitigates hallucinations issues, potentially by alleviating model biases caused by excessive propagation of anchor tokens. Furthermore, we observe a slight reduction in the length of the output sequences generated by MLLMs when using TAME, which may be attributed to the omission of extraneous hallucinated content.

**POPE Evaluation on Hallucinations.** The POPE method evaluates hallucination issues in MLLMs, with a particular focus on object hallucination, similar to CHAIR. POPE employs an essay-

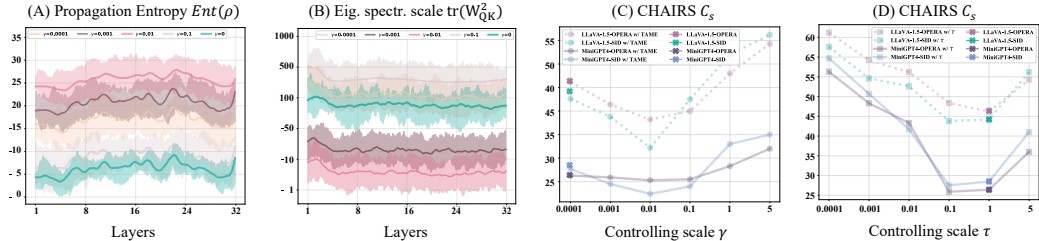

Figure 5: Ablation study of our proposed TAME. (A) Propagation entropy $Ent(\rho)$ under varying $\gamma$. (B) Eigenspectrum scale $Ent(\rho)$ under varying $\gamma$. The shaded areas represent the error bounds at each layer. (C) Comparison of CHAIRS scores between TAME and Baselines across different $\gamma$. (D) Comparison of CHAIRS scores between TAME and Baselines across different $\tau$. In (C), the baselines refer to the models without TAME when $\gamma = 0$, while in (D), it corresponds to $\tau = 1$, where $\tau\sqrt{d} = \sqrt{d}$. (D) demonstrates that $\tau$ is not relevant to mitigating hallucinations.

question format to prompt the model with queries such as "Is there a <object> in the image?" to assess whether the model can accurately identify the presence of a specific object in the image. The evaluation is conducted across three distinct splits: the "random" split, where objects are randomly selected from the entire dataset; the "popular" split, which evaluates the recognition of frequently occurring objects; and the "adversarial" split, which assesses the ability of model to detect objects closely related to those present in the image. We apply the POPE evaluation to four MLLM models, and the average F1 scores are presented in Table 7. While TAME, as a plug-in, effectively enhances performance among the tested decoding strategies. Our approach effectively mitigates hallucinations in both longer sequences and short binary classification tasks by dynamically adjusting token propagation, reducing over-propagation of anchor tokens.

## 6.3 ABLATION STUDY

**Intervening the Eigenspectrum Variance.** We aim to investigate the relationship between the eigenspectrum and token propagation dynamics. Figure 5 (B) illustrates how reparameterization allows us to control the variance of the eigenspectrum. Our empirical analysis shows that the eigenspectrum typically exhibits a large scale, and Figure 5 (A) and (B), indicating that when the eigenspectrum is positioned toward the right of the unimodal distribution of propagation entropy, reducing its scale effectively preserves higher propagation entropy throughout the auto-regressive generation process. As a result, the model demonstrates reduced hallucination rates, as highlighted in Figure 5 (C). This confirms the correlation between the eigenspectrum and hallucinations across various MLLMs, demonstrating that simply adjusting the query-key parameter variance, without incurring additional inference time, can still mitigate hallucinations effectively.

**Temperature $\sqrt{d}$ of Self-Attention.** We set $\tau$ as a control parameter to adjust the temperature $\sqrt{d}$ in attention matrix, *i.e.,* $\tau\sqrt{d}$. This scaling affects the sensitivity of model to inputs. Figure 5 (D) illustrates as $\tau$ increases, CHAIR scores decrease within a range, then slightly rise at higher values, but do not perform better than the original baseline. This indicates that while adjusting $\tau$ influences hallucinations, the effect is not simply due to temperature regulation. Instead, TAME effectively reduces hallucinations at specific $\tau$ values, as shown in Figure 5 (C). In other words, the effectiveness of TAME lies in its dynamic intervention in the eigenspectrum variance of the attention weight matrix, rather than merely adjusting the temperature parameter.

**Case Analysis.** We analyze the attention maps generated by LLaVA1.5-OPERA+TAME and OPERA models to evaluate the impact of TAME on modality alignment in image captioning and visual question answering tasks. Figure 6 illustrates two cases where the attention maps reveal the distribution of attention scores assigned to generated textual tokens within the input image-text sequence during the output generation phase of the MLLM. Our findings show that the OPERA-LLaVA-1.5 model tends to overemphasize the context of the text, which may lead to hallucinations. However, with TAME integrated, the model focuses more on the image, indicating a stronger alignment between image and text modalities. One possible explanation is that by controlling the propagation of anchor tokens and mitigating internal hallucination issues caused by over-propagation, TAME redirects the MLLM's attention, resulting in greater focus on the image tokens.

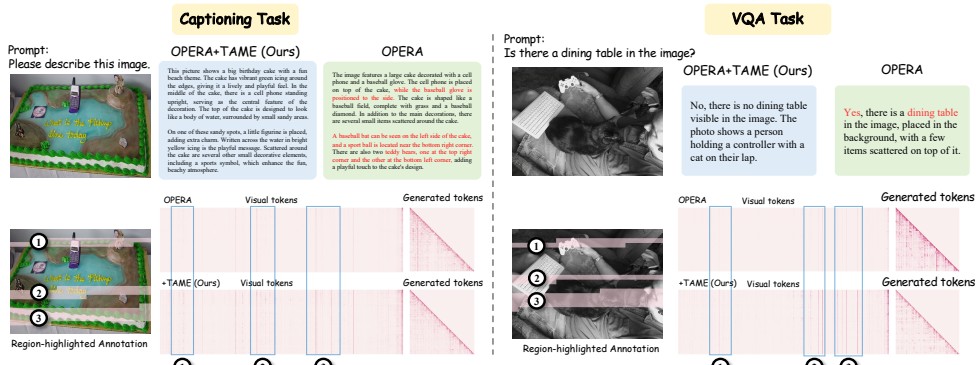

Figure 6: Comparison of attention map between TAME and OPERA-LLaVA-1.5 at different tasks. The blue box region is labeled with the image attentions that can be improved by ours TAME.

# 7 RELATED WORK

**Multimodal Large Language Foundation Models.** Recent advancements in computational resources have significantly boosted research into large-scale foundational models integrated with multi-modal learning (Xu et al., 2024a; Zheng et al., 2024; Hu et al., 2024b;a; Zhang et al., 2024b; Peng et al., 2024b;a). Leveraging open-source large language models like LLaMA (Touvron et al., 2023a;b) and Vicuna (Chiang et al., 2023), MLLMs (Huang et al., 2023; Chen et al., 2023b; Young et al., 2024; Elhoushi et al., 2024; Ye et al., 2024; Li et al., 2024b; Zhang et al., 2024a; Liu et al., 2025a; Wei & Zhang, 2024) can understand and generate a wide range of content more effectively by combining information from multiple modalities, such as text, images, and audio. Models like CLIP and BLIP align text and image features well, while LLaVA (Liu et al., 2024c), InstructBLIP (Dai et al., 2023b)and MiniGPT-4 (Zhu et al., 2023) take this further, enabling users to interact with these systems using images and text prompts. However, they suffer from severe hallucination problems.

**Decoding Strategy in LLMs.** The hallucination (Ji et al., 2023; Liu et al., 2023b; 2024a) in MLLMs refers to the case where the generated text answer does not reflect the true contents of the provided images but rather relies on the internal knowledge of the models. Selecting decoding strategies in language models is crucial, as it determines how models generate text. Top-k sampling decoding (Fan et al., 2018) selects from the top-k most probable tokens, promoting diversity but occasionally resulting in less coherent text. Recent studies propose various decoding strategies (Chuang et al., 2023; Kim et al., 2024; Li et al., 2025; Huang et al., 2024; Wang et al., 2024b; Chen et al., 2024; Leng et al., 2024). They emphasize that hallucinations stem from an over-reliance on anchor tokens, causing subsequent generations to focus more on summarizing the anchor token information rather than utilizing the full context of preceding visual and linguistic tokens. However, these methods lack an analysis of anchor tokens in the decoding strategy and often improve performance at the cost of doubling or even tripling inference time. In this paper, our approach analyzes the relationship between hallucinations and anchor tokens, effectively controlling the generation of hallucination-triggering anchor tokens without incurring additional training, data, or inference time.

# 8 CONCLUSION & LIMITATION

In this paper, we explore the causes of hallucinations in MLLMs by analyzing the propagation patterns of anchor tokens within the attention mechanism. Our findings reveal that over-propagation of anchor tokens, driven by a polarized variance in the eigenspectrum of the QK-parameters, leads to hallucinations by causing the model to neglect visual information. To address this, we propose a plug-and-play decoding strategy, *Dynamic Token Propagation Mechanism* (TAME), which dynamically intervenes the eigenspectrum variance to mitigate excessive propagation. Our approach effectively alleviates hallucinations without complex decoding strategies and incurring extra inference time. Experiments show our superiority in reducing hallucinations on various MLLMs.

**Limitation:** We clarify the limitations of our proposed TAME: *(i):* TAME cannot solve all types of hallucination phenomena in MLLMs. This is understandable because our scheme does not require any additional costs or modifications to the MLLM structure, and therefore has certain limitations in dealing with hallucination problems. *(ii):* Why anchor tokens affect model performance remains elusive. Although the proper propagation of anchor tokens is related to avoiding rank collapse and reducing hallucinations, we need extra effort to fully understand this mechanism in MLLMs.

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

APPENDIX

## A    ATTENTION ALLOCATION

We compute and present the average attention values from the Generated token to both visual tokens and system prompt tokens across different Transformer layers in the pre-trained LLaVA-1.5-7B model. As shown in Figure 7 (light purple trends), attention to visual tokens decreases progressively as the layers deepen. In the shallow layers, the attention distribution is more balanced, with the Generated tokens focusing on both previous output tokens and visual tokens. However, in the deeper layers, the model shifts its focus primarily to the system prompt tokens, reducing attention to visual tokens.

These observations indicate significant redundancy in the visual tokens, particularly in deeper layers, where they contribute less to the output of model. The shift in attention towards the system prompt suggests that as the model processes information through deeper layers, it relies more on the structured prompts than the visual input, revealing potential inefficiencies in visual token utilization.

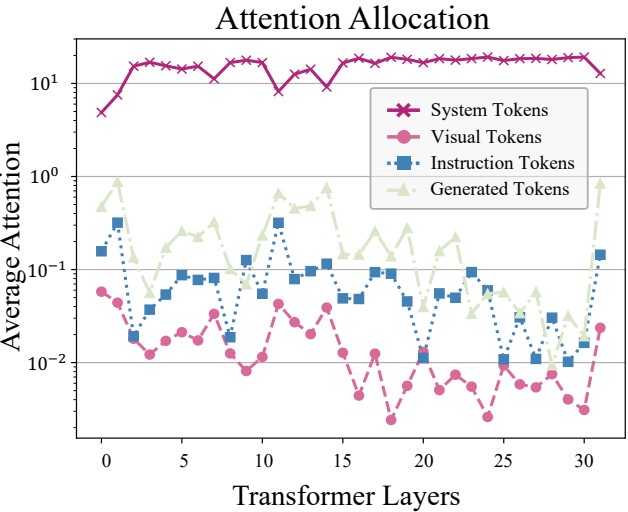

Figure 7: Comparison between the average attention score allocation for different types of token across transformer layers.

## B    DETAILS ABOUT BASELINE

**Sampling-Decoding:** Sampling Decoding generates the next words by randomly selecting from the output probability distribution. Specifically, Top-k sampling (Fan et al., 2018) chooses from the top-k most probable tokens, which introduces diversity into the generated text but can sometimes result in less coherent outputs.

**Greedy-Decoding:** The key distinction between the "Greedy-Decoding" strategy and the "Original" strategy lies in the decoding method used during the generation of image descriptions. In the "Greedy-Decoding" approach, the model opts for greedy decoding instead of sampling, aiming to produce the most deterministic and consistent output. This strategy is employed to examine the possible relationship between the occurrence of hallucinations and the sampling technique.

**Beam-Search: Beam-Search:** Beam Search (Boulanger-Lewandowski et al., 2013; Graves, 2012; Sutskever, 2014) is an advanced decoding technique that maintains a fixed number of hypotheses at each step, allowing for the exploration of multiple potential paths to identify a more optimal sequence. Specifically, with a designated beam size $N_{beam}$, Beam Search retains $N_{beam}$ candidate sequences, each represented by a decoded sequence $x^{N_{beam}}$ accompanied by a corresponding beam score. When generating the token $x_t$, each candidate hypothesis selects $N_{beam}$ possible tokens based

on the top $N_{beam}$ probabilities from the logits. Finally, the decoding process outputs the hypothesis that achieves the highest beam score.

**VCD:** Vision Contrastive Decoding(VCD) manipulates vision inputs by introducing Gaussian noise or directly ablating specific visual features to amplify language priors. By adding Gaussian noise, VCD subtly perturbs the visual data, making the model less confident in relying solely on visual cues and encouraging it to integrate contextual language information more effectively. Direct ablation involves removing or masking certain parts of the visual input, forcing the model to infer missing information based on linguistic context rather than defaulting to learned statistical biases. These manipulations create a contrast between the original and distorted inputs, enabling VCD to identify and suppress tendencies toward hallucination.

**ICD:** Instruction Contrastive Decoding (ICD) methods introduce various types of noisy instructions, including random words, contradictory directives, truncated words, and irrelevant information. For instance, adding random words disrupts instruction coherence, making it difficult for the model to understand the intended meaning. Contradictory directives force the model to depend more on accurate visual inputs instead of relying solely on learned language patterns. Similarly, truncated or incomplete instructions reduce clarity, compelling the model to infer missing information from context.

**OPERA:** Over-trust Penalty and a Retrospection-Allocation strategy (OPERA) penalizes the 'Over-Trust Logit'—a measure of the model's over-reliance on certain tokens—in the beam score. By applying this penalty, OPERA effectively alleviates the aggregation patterns that contribute to hallucinations. This adjustment forces the model to distribute its attention more evenly across relevant tokens, including those representing actual visual content, thereby enhancing the accuracy of the generated descriptions. Although effective, decoding-based methods require iterative decoding, which incurs computational burden and impedes MLLM's deployment on personal devices.

**SID:** Self-Introspective Decoding (SID) is a simple and effective method designed to reduce hallucination issues in Large Vision-Language Models (LVLMs) without relying on additional data, knowledge, or training. SID employs a Context and Text-aware Token Selection (CT2S) strategy, which retains only unimportant visual tokens in the early layers of the model, thereby adaptively enhancing text-based information during the autoregressive decoding process. This strategy ensures that multimodal knowledge guides the model to generate contextually relevant outputs at early stages, rather than aimlessly generating hallucinated content.

## C DETAILS OF EVALUATION

### C.1 DETAILS OF GPT-4 EVALUATION

We generally adopt the GPT-4 evaluation framework introduced in HalluBench (Zhao et al., 2023) and apply it to the VG dataset. Each image in the VG (Krishna et al., 2017) dataset includes comprehensive ground-truth descriptions of all visible objects. Since GPT-4 cannot directly process image data, we incorporate these ground-truth descriptions into the input prompt to help GPT-4 understand the image content. Then, when provided with a description generated by an MLLM in response to the prompt "`Please describe this image in detail,`" GPT-4 is tasked with evaluating whether each sentence in the MLLM's description contains hallucinated information. This evaluation is highly stringent, with GPT-4 marking any descriptions from the MLLM as hallucinations if they differ from the ground-truth details regarding quantity, color, location, activity, or direction.

**Metrics.** There are six metrics considered, which include:

• *The number of sentences per image (SPI).* It reflects the detailedness of MLLM's description at the sentence level.

• *The number of words per image (WPI).* It reflects the detailedness of MLLM's description at the word level.

• *The number of hallucinated sentences per image (HSPI).* It reveals the hallucination degree of MLLM's description at the sentence level. Any sentences that contain hallucinated contents are taken into calculation.

- *The number of hallucinated words per image (HWPI).* It reveals the hallucination degree of MLLM's description at the word level. Any words related with hallucinated contents are taken into calculation.

- *The ratio of hallucinated sentences (HSR).* The average ratio of hallucinated sentences in all sentences of MLLM's descriptions on different images.

- *The ratio of hallucinated words (HWR).* The average ratio of hallucinated words in all words of MLLM's descriptions on different images.

**Prompt.** As shown in Table 2, our adopted GPT-4 prompt is generally based on HalluBench (Zhao et al., 2023).

## C.2 DETAIL OF EVALUATION SCORE

Denoted as $C_S$ and $C_I$, these two variants can be formulated as the average results of

$$C_S = \frac{|\{\text{ hallucinated objects }\}|}{|\{\text{ all mentioned objects }\}|}, C_I = \frac{|\{\text{ captions w/ hallucinated objects }\}|}{|\{\text{ all captions }\}|}$$

where the integration of $\text{CHAIR}_S$ and $\text{CHAIR}_I$ enables a thorough and detailed analysis of object hallucination issues in image captioning.

## C.3 COMPARISON OF TOKEN PROPAGATION PROBABILITY BETWEEN HALLUCINATORY AND NON-HALLUCINATORY CAPTIONS

The objects in this experiment are based on the 80 object labels annotated in (Rohrbach et al., 2018) from the COCO dataset, and the image descriptions are generated by MiniGPT-4 based on inference results from 5000 images in the COCO 2014 train dataset.

## C.4 DETAILS OF GPT-4V EVALUATION

Following (Yin et al., 2023), we perform a dual evaluation on GPT-4V(ision) comparing Beam search decoding with our proposed TAME decoding. Given a trained MLLM model and an image, we generate two descriptions using the prompt "`Please describe this image in detail,`" one with Beam search and the other with TAME. We then use the prompt shown in Table 10 to ask GPT-4V to rate these two descriptions based on the image on a scale from 0 to 10, focusing on two aspects: Accuracy and Detailedness. The accuracy reflects the consistency between the description and the given image. If GPT-4V thinks any content in this description is inconsistent with the given image, namely higher hallucinations, it will get lower score. The detailedness reflects the degree of expressive ability, *i.e.,* how comprehensive does the description characterize the image. The prompt used for GPT-4V, listed in Table 3, instructs it to ignore any bias from the sequential order and to pay special attention to objects mentioned in the MLLM's descriptions that do not appear in the image, including incorrect colors, positions, or relationships. GPT-4V thoroughly analyzes the MLLM's descriptions, leveraging its strong, human-like capabilities.

# D DISCUSSION

## D.1 TOKEN PROPAGATION PROBABILITY AGAINST ATTENTION SCORE

The distinction between Token Propagation Probability and the Attention Score in transformer models is pivotal for understanding token interactions within the self-attention mechanism of large language models (LLMs).

Attention scores, fundamental to the scaled dot-product attention mechanism, measure the similarity between query and key vectors, dictating how much one token should focus on another. These scores are calculated for every token pair and normalized via the softmax function, yielding a probabilistic distribution over attention weights that sum to 1. Although all tokens participate in the attention calculation, those with lower attention weights inherently contribute less to the decisions of model.

On the other hand, token propagation probability estimates the likelihood that a token's information influences the next layer in the network. Unlike attention scores, which do not have an inherent

Table 2: The prompt used for GPT-4 evaluation.

| GPT-4 Prompt |
| --- |
| Please help me judge if the comment of this image is hallucination or correct. I will give you a list of region description of a image. The format is [x1, y1, x2, y2]: region description, where [x1, y1, x2, y2] is the bounding box of the region. Highly overlapping bounding boxes may refer to the same object. This is the ground truth information of the image. Your judgement should base on this information. However, this information only describes the objects in the region of image, so it cannot describe the subjective part of the image, e.g., atmosphere, style, emotion. In that case, you can return "Cannot judge". Also, I will give you a list of comments of the image for you to judge if it is hallucination. Please give a judgement one by one along with the reason. You should pay extra attention to the hallucination, which refers to the part of comments that are inconsistent with the descriptions, specially claiming the existence of something not present in the descriptions. 

 If a comment is hallucination, please help me rewrite it. When rewrite the comment, sound like you are looking at the image directly. Each rewritten comments should compose a description about the image which is correct, detailed, smooth and has strong readability. If not hallucination (correct or cannot judge), keep the original comment. 

 **Your output should be:** 
 **Judgement:** 
 1. hallucination or correct or cannot judge: `<reason>` 
 2. ... 
 **Revised Sentences:** 
 1. ... 
 2. ... 

 **Here are the region descriptions of the image:** 
 {} 
 **Here is the comment for you to judge if it is hallucination and revise:** 
 {} |

threshold, propagation probability introduces a threshold-like behavior by selectively filtering tokens. If a token's activation (or indicator function) is low, it indicates that information of the token is not substantially propagated, effectively discarding less relevant tokens. This metric offers deeper insight into how likely information of a token is to be utilized in subsequent computations, making it an important tool for analyzing cross-layer information flow in LLMs.

While attention scores are computed directly via softmax without any explicit threshold, propagation probability offers a more robust metric for selectively filtering tokens based on their relevance. This mechanism helps mitigate the model's focus on irrelevant tokens, effectively regulating which tokens exert influence across the network. More importantly, token propagation probability not only captures token-wise interactions within a single layer but also models cross-layer information flow, which is critical in autoregressive-based large language models (MLLMs). This provides a theoretical foundation for understanding the phenomenon of over-propagation, a key factor contributing to hallucination in MLLMs

### D.2 COMPARISON OF SUMMARY TOKENS IN OPERA AND ANCHOR TOKENS

We provide clear definitions for Summary Tokens in OPERA and Anchor Tokens below.

**Summary Tokens** are a specific type of token in LLMs that primarily serve to aggregate critical information from a sequence during generation and provide global guidance for subsequent token

Table 3: The prompt used for GPT-4V(ision) evaluation.

---

**GPT-4V(ision) Prompt**

You are required to score the performance of two AI assistants in describing a given image. You should pay extra attention to the hallucination, which refers to the part of descriptions that are inconsistent with the image content, such as claiming the existence of something not present in the image or describing incorrectly in terms of the counts, positions, or colors of objects in the image. Please rate the responses of the assistants on a scale of 1 to 10, where a higher score indicates better performance, according to the following criteria:

1. Accuracy: whether the response is accurate with respect to the image content. Responses with fewer hallucinations should be given higher scores.

2. Detailedness: whether the response is rich in necessary details. Note that hallucinated descriptions should not count as necessary details.

Please output the scores for each criterion, containing only two values indicating the scores for Assistant 1 and 2, respectively. The two scores are separated by a space. Following the scores, please provide an explanation of your evaluation, avoiding any potential bias and ensuring that the order in which the responses were presented does not affect your judgment. `[Assistant 1]`

`{}`
`[End of Assistant 1]`

`[Assistant 2]`
`{}`
`[End of Assistant 2]`

`Output format:`
Accuracy: Scores of the two answers
Reason:

Detailedness: Scores of the two answers
Reason:

---

generation. They reflect the "Aggregation Pattern" inherent to LLMs, enabling the model to synthesize global context for generating coherent outputs. However, excessive reliance on the global information provided by summary tokens may cause the model to overlook original contextual or visual modality inputs, leading to hallucinations.

**Anchor Tokens** are key tokens with high propagation probability in the attention mechanism, especially in multimodal tasks, where they emphasize information interaction between multimodal tokens. However, the over-propagation of anchor tokens can lead to an overemphasis on localized information, causing the generated content to disproportionately focus on specific objects or concepts while neglecting broader contextual cues. This imbalance in attention distribution can contribute to the emergence of hallucinations.

### D.3  ATTENTION MECHANISM

In this section, we discuss concurrent research on understanding the attention mechanism, which is widely used in computer vision (Tang et al., 2024; 2023; Zhao et al., 2024; Xu et al., 2024b; Xiong et al., 2024). (Geshkovski et al., 2024) argued that trained multi-layer self-attention networks exhibit layer-wise dynamics akin to the Kuramoto model, where token embeddings converge to a few "leader" tokens based on the structures determined by the self-attention parameter matrices. (Li et al., 2024c) proved that the learning dynamics of a single-layer self-attention network produce a query-key parameter matrix that captures token-pair frequencies. (Bao et al., 2024) controlled the eigenspectrum variance through regularization dynamics, explicitly steering attention towards localization and thereby preventing the two failure modes of rank collapse and entropy collapse.

By integrating current mainstream MLLM models with the inherent summarization mechanisms of LLMs, we are the first work to explore the promising direction of studying the implicit bias of attention through parameter eigenspectra. We discuss the strong correlation between over-propagation and hallucinations, highlighting that reducing the propagation probability of anchor tokens can effectively alleviate the hallucination problem. Our method, TAME, addresses over-propagation by intervening in the eigenspectrum of the query-key parameter matrix, without requiring additional training or inference time.

### D.4 SOCIETAL IMPACTS

TAME does not pose any potential social harm. On the contrary, it has the potential to significantly advance the development of multimodal large language models (MLLMs). TAME provides inspiration for the research community, encouraging the exploration of more efficient solutions to mitigate the hallucination issue in MLLMs without incurring additional costs. It not only avoids any negative social impact but also promotes the progress of multimodal AI assistants. These methods can achieve better generalization across different types of MLLMs. Currently, although MLLMs rely on large language models, they still lack modules that resemble human brain functions, which need to be developed at the architectural level.

## E IMPLEMENTATION DETAILS

Basically, the hyperparameter gamma of TAME is set to the default value of 1. Other parameters use the default settings, same as the Baseline. Experiments are performed on NVIDIA H20/H100 GPUs.

To accelerate computation, we adopt the power method to approximate the eigenspectrum variance of the current matrix, with Algorithm 1 providing a brief implementation sketch. In practice, fp32 precision is typically required to ensure numerical stability. We experimented with applying the TAME method as a plug-and-play approach to key and query weights, and confirmed its robustness across different configurations. Applying TAME to all layers is the simplest and most effective solution, performing well in practice without introducing any additional overhead.

---

**Algorithm 1** Pseudo-code of TAME in a PyTorch-like style.

---

```
# Parameters:
# W: Weight matrix of shape (bs, h, d, c)
# gamma: Hyperparameter , shape (,1)

# Initialize gamma as a tunable hyperparameter, with a default value of 1

# Compute the trace of W^2 eta using Monte Carlo estimation
trace_squared_estimates = []

for i in range(num_samples):
    idx = random_integer(0, W.size(0)) # Randomly select a row/column index
    estimate = (W[idx, idx])^2 # Take the diagonal element and square it
    trace_squared_estimates.append(estimate)

eta = sum(trace_squared_estimates) / num_samples # Average the estimates

# Dynamically intervening W according to the computed trace of W^2
W_hat = (1 + gamma / log(eta + xi)) * W

# Return reparameterized weight matrix W_hat
return W_hat
```

---

## F MORE RESULTS

### F.1 EVALUATION ON CHALLENGING BENCHMARKS

We conduct a rigorous evaluation of VCD, OPERA, and RAG (Vanilla-RAG (Karpukhin et al., 2020), SURf-RAG (Sun et al., 2024)) as well as RLHF (RLHF-V (Yu et al., 2024), CSR (Zhou et al., 2024)) on six popular MLLM benchmarks and four additional ones, as shown in the table 4.

For details, **SEED-Bench** (Li et al., 2023a) consists of 19k multiple choice questions with human annotations, while spanning 12 evaluation dimensions. **GQA** (Hudson & Manning, 2019) incorporates a novel evaluation metrics suite focused on consistency, grounding, and plausibility, establishing a rigorous standard for assessing in vision-language tasks. **Vizwiz** (Gurari et al., 2018) examines certain perception capability, like knowledge and relation. **MME** (Fu et al., 2023) contains 14 meticulously designed subtasks that challenge the models' interpretative and analytical skills. **MM-Bench** (Liu et al., 2025b) also examines LVLMs on general perception capabilities using a wide range of tasks. **POPE** is an assessment methodology designed to scrutinize object hallucination in LVLMs.

**Vanilla-RAG** concatenates the Top-N image-caption pairs from the database, which have the highest CLIP score similarity to the test image, before appending the questions and images for the LVLMs to respond. **SURf-RAG** is a self-refinement framework that teaches LVLMs to selectively utilize retrieved information. **RLHF-V** collects fine-grained paragraph-level corrections from humans on hallucinations and performs dense direct preference optimization using human feedback. **CSR** enables the model to self-improve by iteratively generating candidate responses, evaluating the reward for each response, and curating preference data for fine-tuning.

For the RAG and RLHF methods, our approach seamlessly integrates as a plug-and-play module within their frameworks, without incurring extra inference time. By combining RAG with TAME, external knowledge retrieval enhances the accuracy of generated outputs, ensuring better alignment with the input context. Similarly, when integrated with RLHF, our method leverages human feedback to guide the model in producing outputs that are more faithful to the actual content. Experimental results demonstrate that our approach delivers comprehensive improvements across both frameworks, further validating its effectiveness.

We conduct additional experiments to evaluate improvements in factuality metrics and the effectiveness of data-driven instruction tuning on model performance. Our evaluations focus on benchmarks such as **GQA**, **VizWiz**, **MME**, and **POPE**, which test real-world knowledge QA and multimodal understanding tasks, as shown in the table below. The results demonstrate that instruction tuning with LRV-Instruction-finetuned (Liu et al., 2023b) moderately enhances performance by leveraging high-quality image-text pairs for task-specific fine-tuning, improving the model's alignment with real-world knowledge. However, integrating TAME further amplifies these gains by dynamically mitigating hallucinations and strengthening factual alignment, resulting in significant improvements across all benchmarks. This combination achieves a higher degree of correctness and consistency in generated outputs, validating that TAME effectively complements instruction tuning as a plug-and-play enhancement, improving both accuracy and robustness in real-world multimodal tasks.

### F.2    EXTENDING TAME TO SINGLE-MODAL LLMS

We extend our experiments to single-modal LLMs. As shown in table 5, we conduct on the Wikitext-103 (Merity et al., 2016) and MiniPile (Kaddour, 2023) datasets, to assess the scalability and consistency of TAME's impact. TAME was integrated as a plug-and-play enhancement across three distinct model configurations, including the BLOOM (LLaMA architecture with ALiBi) (Lester et al., 2021) and OpenLLaMA (Touvron et al., 2023a). The results showed that TAME consistently improved perplexity (PPL) across all architectures and parameter sizes, demonstrating its effectiveness in enhancing model performance. These findings further support that TAME is a generalizable mechanism, suitable for both multimodal and single-modal LLMs, broadening its applicability.

### F.3    GPT-4V ASSISTED EVALUATION

We further resort to GPT-4Vision, a strong multi-modal assistant that can easily handle the input from vision, language, and voice modality. Typically, we randomly sample 500 images from MSCOCO's validate set and ask different MLLM models to describe these images. For fair comparison, we follow (Yin et al., 2023) and compare the answers obtained from two decoding methods at the same time, *i.e.,* providing the image and both the answers to GPT-4V and prompting it to give a judgement from 0-10 for each. The prompt emphasizes mitigating the impact of the sequential order fed to GPT-4V and, additionally, paying special attention to the objects mentioned in answers but not appear in the image. It includes instances where the objects are represented in an incorrect form of colors, positions, or relationships.

| Methods | GQA | SEED[I] | VisWiz[V] | MME[O] | MME[A] | MMB | POPE[R] |
|---|---|---|---|---|---|---|---|
| **LLaVA-1.5-7B** | 60.4 | 58.1 | 49.0 | 278.33 | 245.00 | 64.2 | 80.7 |
| + VCD | 61.0 | 58.9 | 50.8 | 293.00 | 268.33 | 61.4 | 83.3 |
| + VCD w/ **TAME** | 61.7 | 59.4 | 51.6 | **295.67** | **275.67** | 61.5 | 84.0 |
| + OPERA | 62.0 | 59.6 | 52.4 | 290.33 | 251.67 | 64.8 | 83.4 |
| + OPERA w/ **TAME** | 62.5 | **60.7** | 52.9 | 294.33 | 256.00 | 65.3 | 83.8 |
| + Vanilla-RAG | 60.2 | - | 49.6 | 264.33 | 255.33 | - | 84.7 |
| + Vanilla-RAG w/ **TAME** | 61.8 | - | 50.7 | 272.33 | 259.67 | - | 86.5 |
| + SURf-RAG | 62.4 | - | 54.3 | 268.67 | 253.00 | - | 85.6 |
| + SURf-RAG w/ **TAME** | **62.9** | - | **55.2** | 270.00 | 258.33 | - | 87.1 |
| + RLHF-V | 62.3 | 59.3 | 53.7 | 283.33 | 263.67 | 63.6 | 85.2 |
| + RLHF-V w/ **TAME** | 62.8 | 59.8 | 54.2 | 272.33 | 266.33 | 64.2 | 86.1 |
| + CSR | 61.9 | 60.0 | 53.4 | 285.67 | 264.67 | 65.2 | 86.0 |
| + CSR w/ **TAME** | 62.4 | 60.6 | 53.9 | 294.00 | 271.00 | **66.9** | **87.0** |
| **LLaVA-1.5-13B** | 64.1 | - | 53.3 | - | - | 68.2 | 86.5 |
| + OPERA | 64.0 | - | 55.6 | - | - | **68.9** | 87.2 |
| + OPERA w/ **TAME** | **65.7** | - | **56.2** | - | - | 68.8 | **89.0** |
| **LLaVA-NeXT-Mistral-7B** | 62.9 | - | 52.6 | - | - | 66.1 | 88.2 |
| + OPERA | 63.4 | - | 52.9 | - | - | 67.2 | 88.7 |
| + OPERA w/ **TAME** | **64.6** | - | **54.0** | - | - | **67.4** | **90.4** |
| **mPLUG-Owl-7B** | 66.7 | 63.5 | 57.1 | 310.33 | 281.67 | 68.2 | 89.1 |
| + LRV-Instruction-finetuned | 67.5 | 64.2 | 58.0 | 315.00 | 285.33 | 69.5 | 90.0 |
| + LRV-Instruction-finetuned w/ **TAME** | **68.9** | **65.8** | **59.2** | **322.33** | **291.00** | **71.0** | **91.5** |

Table 4: Comparison of methods on different benchmarks. **SEED**[I] refers to SEED *image* evaluation, **VisWiz**[V] refers to *image* refers to VisWiz VQA, **POPE**[R] refers to POPE Random, **MME**[O] refers to MME Object-level Hallucination Existence Count.

| WikiText-103 | | | | MiniPile | | |
|---|---|---|---|---|---|---|
| Model | #Params | PPL | | Model | #Params | PPL |
| BLOOM | 71M | 29.9 | | BLOOM | 160M | 25.8 |
| BLOOM w/ **TAME** | 71M | **29.0** | | BLOOM w/ **TAME** | 160M | **25.3** |
| OpenLLaMA | 71M | 27.4 | | OpenLLaMA | 160M | 25.9 |
| OpenLLaMA w/ **TAME** | 71M | **26.9** | | OpenLLaMA w/ **TAME** | 160M | **24.9** |
| BLOOM | 160M | 27.6 | | BLOOM | 430M | 20.6 |
| BLOOM w/ **TAME** | 160M | **26.0** | | BLOOM w/ **TAME** | 430M | **19.3** |
| OpenLLaMA | 160M | 22.5 | | OpenLLaMA | 430M | 19.6 |
| OpenLLaMA w/ **TAME** | 160M | **21.3** | | OpenLLaMA w/ **TAME** | 430M | **19.4** |

Table 5: Perplexity (PPL) comparison on WikiText-103 and MiniPile datasets using BLOOM and OpenLLaMA architectures with and without TAME across varying parameter sizes.

As demonstrated in Table 6, TAME delivers improvements of up to 25.7% compared to Beam Search decoding and enhances performance over state-of-the-art methods by as much as 6.8%, all while maintaining the level of detail in responses. Given that GPT-4V's perceptual and reasoning capabilities closely approximate human judgment, the evaluation results from GPT-4V provide a strong indication the effectiveness of TAME in reducing hallucinations from a human-centric perspective.

## F.4 EVALUATING ON DIVERSE HALLUCINATION TYPES

The table 8 presents our experimental results on the hallucination subset of the MME dataset, which includes object-level hallucinations and attribute-level hallucinations. By comparing the performance of various methods, we demonstrate the broad applicability of TAME. As a plug-and-play decoding strategy, TAME achieves significant performance improvements across multiple methods. By dynamically adjusting the eigenspectrum variance of the attention weight matrix, TAME effectively mitigates the over-propagation of anchor tokens, thereby enhancing model performance

Table 6: GPT-4V assisted hallucination evaluation (Huang et al., 2024) results on MSCOCO. Two aspects are verified, *i.e.,* correctness (*C*) and detailedness (*D*). Higher correctness/detailedness indicates less hallucinations.

| Methods | InstructBLIP | | MiniGPT-4 | | LLaVA-1.5 | |
|---|---|---|---|---|---|---|
| | *C* | *D* | *C* | *D* | *C* | *D* |
| Beam | 5.68 | 5.35 | 5.47 | 5.16 | 5.53 | 5.55 |
| **+TAME** | **6.24** | **5.78** | **6.68** | **5.20** | **6.39** | **5.82** |
| OPERA | 6.19 | 5.67 | 6.62 | 5.18 | 6.23 | 5.79 |
| **+TAME** | **6.45** | **5.80** | **6.84** | **5.46** | **6.37** | **5.80** |
| SID | 6.26 | 5.73 | 6.54 | 5.21 | 6.25 | 5.84 |
| **+TAME** | **6.47** | **5.78** | **6.89** | **5.24** | **6.42** | **5.95** |

Table 7: POPE hallucination evaluation results for four MLLM models, presenting the average F1-score calculated across the *random*, *popular*, and *adversarial* splits of POPE.

| Methods | InstructBLIP | MiniGPT-4 | LLaVA-1.5 | Shikra |
|---|---|---|---|---|
| Beam | 77.9 | 75.1 | 80.8 | 80.1 |
| **+TAME** | **80.1** | **78.1** | **82.6** | **81.9** |
| VCD | 81.9 | 74.7 | 83.3 | 82.5 |
| **+TAME** | **84.5** | **79.4** | **84.0** | **83.7** |
| ICD | 82.3 | 74.4 | 82.2 | 81.7 |
| **+TAME** | **83.4** | **79.8** | **83.1** | **82.6** |
| OPERA | 82.7 | 76.6 | 83.4 | 81.8 |
| **+TAME** | **84.9** | **79.2** | **83.8** | **82.3** |
| SID | 82.2 | 77.0 | 83.5 | 82.2 |
| **+TAME** | **82.5** | **79.3** | **84.1** | **83.9** |

across multiple dimensions, including object existence, count estimation, position alignment, and color consistency.

| Decoding | Object-level | | Attribute-level | | Total Scores↑ |
|---|---|---|---|---|---|
| | *Existence*↑ | *Count*↑ | *Position*↑ | *Color*↑ | |
| **LLaVA-1.5-7B** | 163.67 | 114.66 | 104.00 | 141.00 | 523.33 |
| + Greedy | 184.00 | 95.33 | 112.00 | 157.67 | 549.00 |
| + VCD | 172.67 | 120.33 | 129.67 | 155.00 | 561.33 |
| + OPERA | 174.67 | 115.66 | 110.67 | 141.00 | 542.00 |
| + OPERA w/ **TAME** | 176.00 | 118.33 | 113.00 | 143.00 | 550.33 |
| + SID | 182.00 | 127.00 | 116.00 | 139.00 | 564.00 |
| + SID w/ **TAME** | **193.00** | **137.33** | **139.00** | **164.67** | **634.00** |

Table 8: Evaluation results on the hallucination subset of MME. max-tokens=512.

## F.5 LAYER-WISE EVALUATION OF TAME: EXPLORING ITS IMPACT ON TOKEN PROPAGATION

As analyzed in Figure 1 and Appendix A, early, middle, and late layers exhibit significant differences in token propagation patterns. Early layers ($l_0$) primarily focus on extracting low-level features and token initialization, middle layers ($l_{16}$) emphasize multi-modal alignment and feature aggregation, while late layers ($l_{31}$) are responsible for high-level reasoning and final output generation. These differences play a critical role in the model's behavior and error generation, especially in hallucination-prone scenarios.

We evaluate TAME by applying it to different combinations of layers ($l_0$, $l_{16}$, and $l_{31}$), as shown in the table 9. In this experiment, the baseline is OPERA (Exp I). The results from Exp II-IV demonstrate the incremental impact of TAME when applied to individual layers: **Early layers ($l_0$, Exp II)**: Applying TAME at this stage slightly reduces hallucinations by refining token initialization and propagation, but its overall impact is limited due to the lack of deeper-layer information. **Middle layers ($l_{16}$, Exp III)**: Incorporating TAME at this stage significantly improves multi-modal alignment, optimizing feature integration and enhancing response generation accuracy. **Late layers ($l_{31}$, Exp IV)**: Applying TAME here substantially reduces errors, as this stage directly influences final reasoning and output generation.

The results from Exp V-VIII further demonstrate the cumulative effect of applying TAME to multiple layers: $l_0 + l_{16} + l_{31}$ (**Exp VIII**): Applying TAME across all layers achieves the best overall performance, with the fewest hallucinations and the highest accuracy across all tasks, albeit at a slightly higher inference cost.

This phenomenon highlights the importance of holistically optimizing token propagation across the model. Early layers provide foundational improvements, middle layers optimize multi-modal representations, and late layers ensure high-level reasoning accuracy. The experiments suggest that integrating TAME across more layers significantly reduces error-prone responses, as it dynamically mitigates over-propagation at different stages of the model.

| Exp | $l_0$ | $l_{16}$ | $l_{31}$ | MMB | GQA | CHAIR$_S$ | CHAIR$_I$ |
|---|---|---|---|---|---|---|---|
| I | | | | 64.8 | 62.0 | 46.4 | 13.0 |
| II | ✓ | | | 64.9 | 62.0 | 46.2 | 12.7 |
| III | | ✓ | | 64.8 | 62.1 | 45.8 | 12.6 |
| IV | | | ✓ | 64.9 | 62.3 | 45.9 | 12.7 |
| V | ✓ | ✓ | | 65.1 | 62.3 | 44.2 | 12.6 |
| VI | ✓ | | ✓ | **65.3** | **62.6** | 44.4 | 12.5 |
| VII | | ✓ | ✓ | 65.2 | 62.4 | 43.9 | 12.3 |
| VIII | ✓ | ✓ | ✓ | **65.3** | 62.5 | **41.3** | **12.2** |

Table 9: Ablation study of applying TAME on different layers. $l_0$ : layer 0; $l_{16}$ : layer 16; $l_{31}$ : layer 31;

## F.6 COMPARISON OF OUTPUT TEXT LENGTH ACROSS DECODING METHODS

We conduct detailed evaluations of the generated output length across different decoding methods, as shown in the table 10. The results indicate that while TAME effectively reduces hallucinated content, it maintains or even slightly increases the output length when integrated with decoding strategies like VCD, OPERA, or ICD. Specifically, on the COCO dataset, TAME achieves a balanced reduction in hallucinations without sacrificing detail richness, as demonstrated by the minimal variation or slight increase in the average length.

| Method | Length |
|---|---|
| LLaVA-1.5 | 100.6 |
| + VCD | 100.4 |
| + VCD w/ **TAME** | 100.9 |
| + OPERA | 98.6 |
| + OPERA w/ **TAME** | 98.4 |
| + ICD | 106.3 |
| + ICD w/ **TAME** | 110.1 |

Table 10: Comparison of the hallucination mitigation performance across different methods in terms of output length.

## F.7 TEXT QUALITY EVALUATION.

To assess the overall quality of generated text comprehensively, we adopt PPL (Perplexity, a classical metric in NLP without using reference text), and resort to GPT-4 to assess the grammar, fluency, and naturalness of generated text. We randomly select 1,000 images in MSCOCO and verify on LLaVA-1.5 7B model. The average results are listed above, where PPL1 and PPL2 are calculated by pretrained gpt2 and gpt2-medium respectively. From the results in Table 11, we discover that TAME can generally keep the quality of generated text from various aspects. Besides, we test TAME on two popular MLLM benchmark, *i.e.,* MME and MMBench (Liu et al., 2023c), using LLaVA-1.5 7B model. Table 12 shows that TAME can maintain and even improve MLLM's performance on both MLLM benchmarks.

Table 11: The evaluation results for the quality of generated text. We calculate $PPL_1$ and $PPL_2$ with gpt2 and gpt2-medium in the Huggingface model zoo, respectively. The ratings of grammar, fluency, and naturalness are provided by GPT-4.

|  | $PPL_1 \downarrow$ | $PPL_2 \downarrow$ | Grammar $\uparrow$ | Fluency $\uparrow$ | Natural $\uparrow$ |
|---|---|---|---|---|---|
| Greedy | 12.72 | 10.27 | 9.58 | 9.01 | 8.52 |
| **+TAME** | 12.63 | 10.04 | 9.59 | 9.12 | 8.57 |
| Beam Search | 11.11 | 8.89 | 9.54 | 8.95 | 8.55 |
| **+TAME** | 11.04 | 8.79 | 9.58 | 8.98 | 8.56 |
| OPERA | 11.67 | 9.31 | 9.54 | 8.93 | 8.53 |
| **+TAME** | 11.56 | 9.17 | 9.60 | 8.98 | 8.52 |
| SID | 11.70 | 9.35 | 9.49 | 9.06 | 8.47 |
| **+TAME** | 11.62 | 9.30 | 9.55 | 9.09 | 8.49 |

Table 12: TAME generally improves the MLLM's performance on popular MLLM benchmark.

|  | Greedy | Beam | OPERA | **OPERA + TAME** |
|---|---|---|---|---|
| MMBench | 64.3 | 64.4 | 64.4 | 65.2 |
| MME | 1510.7 | 1504.3 | 1515.4 | 1523.0 |

# G  DETAILED PROOFS

**Theorem 1** (Propagation Entropy). *Let* $\sigma = \|W_K W_Q^\top\|_2 \|XX^\top\|_2$, *and* $\beta = \exp\left(-\sigma\sqrt{\frac{T}{T-1}}\right)$. *The propagation entropy* $Ent(\rho)$ *holds that:*

$$Ent(\rho) = \sigma \log(1 + (T-1)\beta) + \frac{\sigma^2 \sqrt{T(T-1)}\beta}{1 + (T-1)\beta},$$

*where* $Ent(\rho)$ *represents that lower entropy increases the likelihood of over-propagation of anchor tokens, following a unimodal pattern in* $\sigma$, *and vanishing as* $\|W_{QK}\|_2 \to 0$ *or* $\infty$ *as illustrated in Figure 3(Left). Propagation entropy increases with* $\|W_{QK}\|_2$ *up to a peak, then decreases, reaching its lowest point at extreme values of* $\|W_{QK}\|_2$. *If* $|\operatorname{tr}(W)|$ *is moderate, propagation entropy stays near the peak. To mitigate over-propagation of anchor tokens, it is sufficient to control* $\operatorname{tr}(W^2)$ *under a fixed* $\operatorname{tr}(W)$: $\left\|\Sigma^{-1}\right\|_F \sqrt{\operatorname{tr}(W^2)} \geq \|W_{QK}\|_2$.

*Proof.* Let $a \in \mathbb{R}^T$ denote the $i$'th row of $A$, $a = A_i$. From the assumptions it holds that $\|a\| \leq \sigma$. $\rho = \rho(a)$ in Eq. 1 denote the softmax propagation probabilities given by:

$$\rho_i = \frac{e^{a_i}}{\sum_{k=1}^T e^{a_k}},$$

The entropy still follows the classical probability-based definition. For the new $\rho_i$, propagation entropy can be written as:

$$Ent(a) = -\sum_{i=1}^T \rho_i \log \rho_i = -\sum_{i=1}^T \frac{e^{a_i}}{Z} \log\left(\frac{e^{a_i}}{Z}\right).$$

We are minimizing the entropy based on $\rho_i$, while subject to the quadratic constraint on $a$:

$$\min_a Ent(a) \quad \text{s.t.} \quad \|a\|^2 \leq \sigma^2.$$

The Lagrangian function is defined to handle the constraint:

$$\mathcal{L}(a,\lambda) = Ent(a) + \frac{1}{2}\lambda(\|a\|^2 - \sigma^2).$$

To find all saddle points, we solve the system of equations:

$$\frac{\partial \mathcal{L}(u,\lambda)}{\partial u} = 0, \quad \frac{\partial \mathcal{L}(u,\lambda)}{\partial \lambda} = 0$$

Based on the final equation shown and your reference to the work of (Zhai et al., 2023), the propagation entropy is derived and represented as:

$$\text{Ent}(a^\star) = \log\left(1 + (T-1)e^{-\sigma\sqrt{\frac{T}{T-1}}}\right) + \frac{\sigma\sqrt{T(T-1)}e^{-\sigma\sqrt{\frac{T}{T-1}}}}{1 + (T-1)e^{-\sigma\sqrt{\frac{T}{T-1}}}}.$$

To maintain a degree of localization and avoid rank collapse, as discussed in Eq.9, we added a constraint to $Ent(a^\star)$, such that $Ent^{'}(a^\star) = \sigma Ent(a^\star)$. Therefore, the final propagation entropy is given by:

$$Ent^{'}(a^\star) = \sigma \log\left(1 + (T-1)\beta\right) + \frac{\sigma^2\sqrt{T(T-1)}\beta}{1 + (T-1)\beta},$$

where $\beta = \exp\left(-\sigma\sqrt{\frac{T}{T-1}}\right)$. The first inequality is due to the Cauchy–Schwarz inequality as:

$$\|\Sigma^{-1}\|_F \sqrt{\operatorname{tr}(W^2)} = \|\Sigma^{-1}\|_F \|W\|_F \geq \|W_{QK}\|_F \geq \|W_{QK}\|_2,$$

$\square$

## H   VISUALIZATION

In this section, we present a visualization that illustrates the relationship between the attention map, token propagation probability, and the generated tokens. The visualization reveals that tokens with exceptionally high propagation probabilities trigger a series of hallucinations in the subsequent generated text.

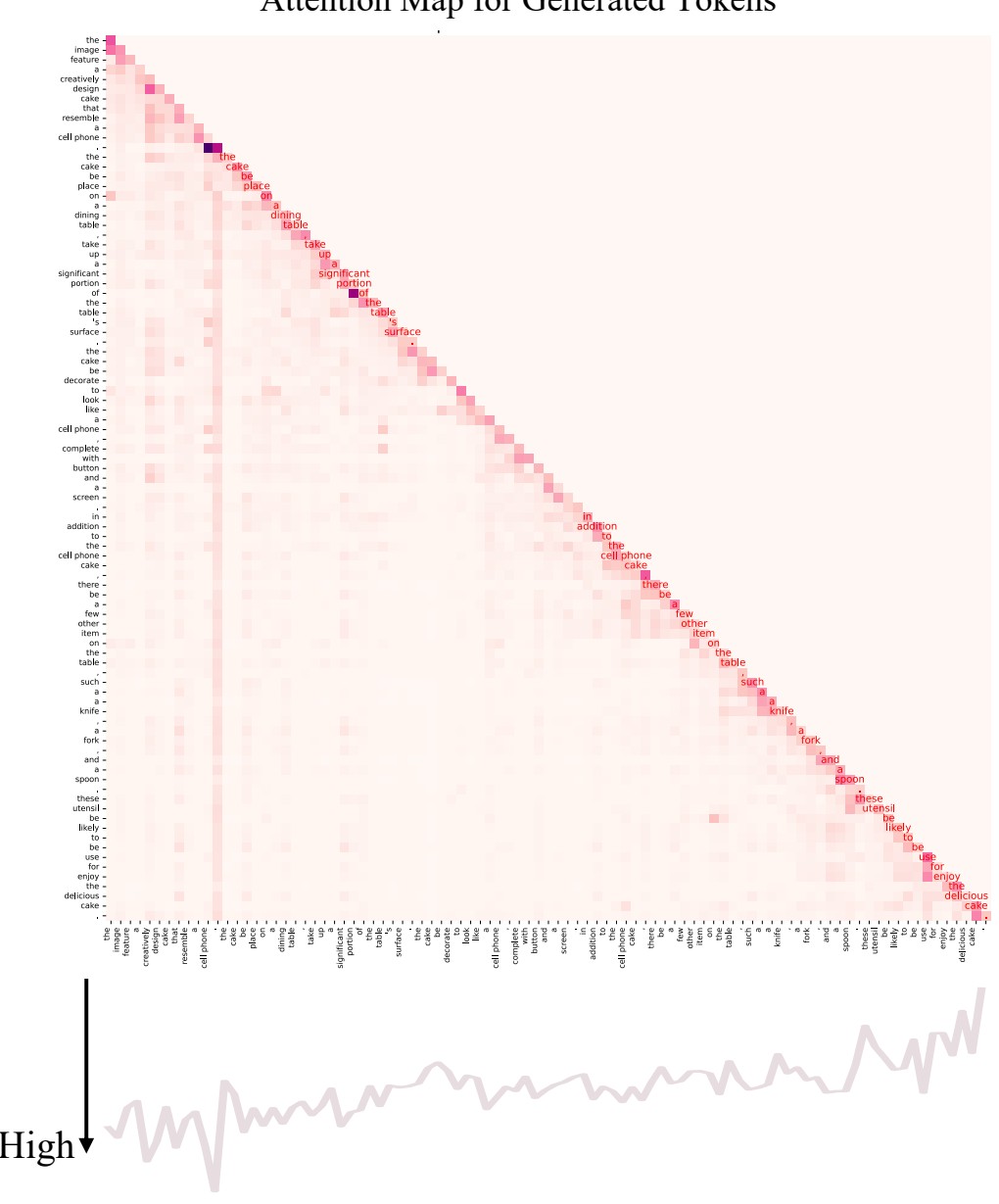

Figure 8: Visualization of attention map and token propagation probability for generated tokens. The words in hallucinated captions are marked in red.

## I   MORE QUALITATIVE RESULTS

In this section, we provide additional visual comparisons to further demonstrate the effectiveness of TAME. These visualizations offer clear evidence of how TAME improves the model's performance

**Instruction:** Please describe this image in detail

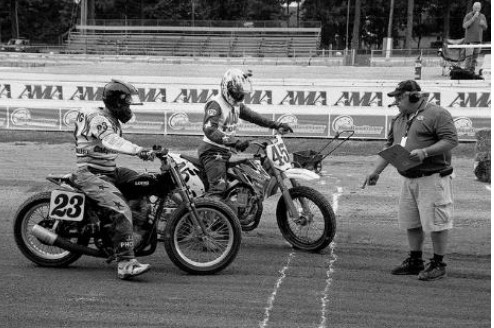

**Input Image**

*Shikra (Beam Search):*

The image feature a black and white scene of several motorcycle park next to each other on a dirt road . the motorcycle be arrange in a row , with one occupy the majority of the frame on the left side and the others gradually decrease in size . there be three motorcycle in total , with one prominently position in the middle and the other on the right side . in the background , there be a few chair and a bench scatter around. the chair be place in various position , include one in the left foreground , another in the middle , and the third towards the side . the bench be locate on the right side of the scene , near trighthe motorcycle . additionally , there be a person stand on the far left side , close to the chair .

*Shikra (OPERA):*

The scene feature a group of people on dirt bike , line up next to each other and prepare for a race . three men can be see prominently on their motorcycle , with one person position close to the front and the other two on either side . there be two more people , one near the center and another on the left side of the scene . the motorcycle be arrange in a row , with one bike near the center and the other two towards the right side of the scene . A chair be also visible in the middle of the scene , possibly for rest or for the race official .

*Shikra (Beam Search+TAME):*

The image feature a red train with the word `` howrah '' write on it side , park at a train station . the train be position on the track , occupy a significant portion of the scene . the train station appear to be a busy area , with multiple people scatter throughout the scene , likely wait for their train or attend to other task . the presence of several individual suggest that the train station be a bustle hub of activity .

*Shikra (OPERA+TAME):*

The image depict a group of three men rid motorcycle on a dirt track . they be all wear helmet and appear to be participate in a race . the motorcycle be position closely together , with one on the left , another in the middle , and the third on .

**Instruction:** Please describe this image in detail

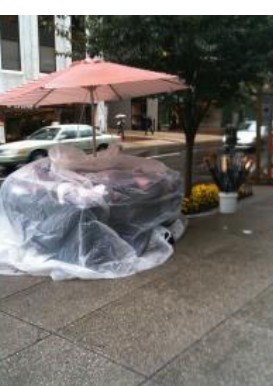

**Input Image**

*Shikra (Beam bearch):*

The image feature a large umbrella cover a car park on the side of a street . the umbrella be position above the car , provide protection from the element . the car be park next to a building , and there be several other car park in the vicinity . in addition to the car , there be multiple people walk around the area . some of them be carry handbag , with one person hold a handbag near the center of the scene and another person carry a handbag closer to the right side . the presence of pedestrian and the parked car create a lively urban atmosphere .

*Shikra (Opera):*

The image depict a street scene with a large umbrella cover a couch place on the sidewalk . the umbrella be position above the couch , provide shade and protection from the element . the couch be surround by several potted plant , add a touch of greenery to the

*Shikra (Beam Search+TAME):*

The image feature a car cover in plastic , likely to protect it from the element . the car be park on the side of the street , and it appear to be a convertible . the car be park next to a building , and there be a person stand nearby . the scene also include a few other car park in the background .

*Shikra (Opera+TAME):*

The scene feature an umbrella cover a black car park on the side of a busy city street . the umbrella be position above the car , protect it from the rain . there be several other car park in the vicinity , include one close to the right edge of the image .

Figure 9: Qualitative example.

by reducing hallucinations and enhancing attention localization. As shown in Figure 9, Figure 10, Figure 11, and Figure 8, our proposed TAME not only mitigates hallucinations in large language models but also ensures that the generated text remains of high quality. By comparing the outputs of models with and without TAME, we highlight the significant differences in how the attention mechanism processes input tokens and distributes weights across the sequence.

å

*Instruction:* Please describe this image in detail

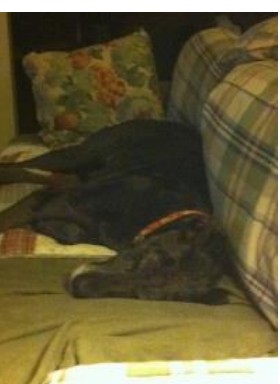

**Input Image**

### *Shikra (Beam Search):*
the image feature a large , fluffy dog lie on a bed , partially underneath a pillow . the dog occupy the majority of the scene , with it body position across the width and length of the bed . the bed appear to be cover in blanket and pillow , give the dog a cozy and comfortable spot to rest . the overall scene be warm and inviting , with the dog enjoy a relaxing moment in it bed .

### *Shikra (OPERA):*
the image feature a large , elderly dog lie on a bed with it head rest on a pillow . the bed be cover in a plaid blanket , give it a cozy and comfortable appearance . the dog occupy a significant portion of the bed , extend from one end to the other and almost reach the edge of the frame .

### *Shikra (Beam Search+TAME):*
the image feature a large black dog lay on a couch , rest it head on a pillow . the dog be position in the middle of the couch , occupy a significant portion of the space . there be two pillow on the couch , one on the left side and another on the

### *Shikra (OPERA+TAME):*
the image depict a black dog sleep comfortably on a couch . the dog occupy a large portion of the couch , with it head rest on a pillow . the couch have a plaid pattern , add texture to the scene . in the background , there be two pillow position , one behind the dog 's head and another on the right side of the couch . the cozy arrangement of the pillow and couch provide a relaxing environment for the dog .

---

*Instruction:* Please describe this image in detail

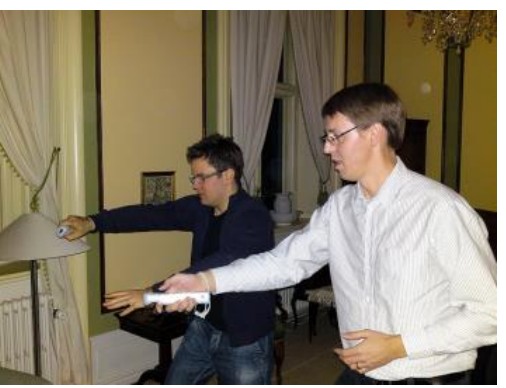

**Input Image**

### *Shikra (Beam Search):*
the image show two men play a video game in a dimly lit room , each hold a nintendo wii controller and actively engage in the game . one of the men appear to be wear glass , and they both seem to be enjoy themselves a they move their controller and compete . in the room , there be a chair and a couch , as well a a potted plant , a bowl , and a cup place in the background . the setting create a cozy atmosphere for the two friend to enjoy their gaming session together .

### *Shikra (OPERA):*
the image depict two young men stand in a living room , play a video game on a nintendo wii console . both men be hold wii remote in their hand , fully immerse in the game . they be position close to each other , with one man on the left and the other on the right side of the room . the living room be furnish with a couch in the background and a chair nearby . there be two potted plant in the room , one on the left side and the other on the right side , add a touch of greenery to the space . a vase can also be see on the right side of the room , further enhance the room 's decor .

### *Shikra (Beam Search+TAME):*
the image depict two young men stand in a living room , play a video game on a nintendo wii console . both men be hold wii remote in their hand , fully engage in the game . one of the men be wear glass .

### *Shikra (OPERA+TAME):*
the image feature two young men stand in a living room , play a video game together . both men be hold wii controller , actively engage in the game . they be position close to each other , with one man on the left side and the other on the right side of the room .

Figure 10: Qualitative example.

*Instruction:* Please describe this image in detail

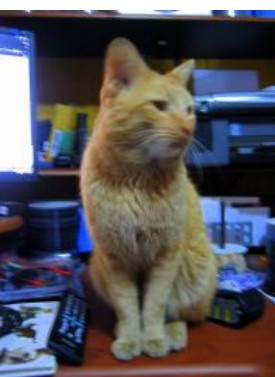

**Input Image**

*Shikra (Beam Search):*

the image feature a yellow cat sit on a desk , position in front of a monitor . the cat occupy a large portion of the desk , and it orange fur stand out against the wooden surface . on the desk , there be several item , include a keyboard , a mouse , and two cell phone . additionally , there be multiple book scatter around the desk in various orientation , give the scene a cluttered appearance . a chair be situate near the desk , and a tv can be see in the background , add to the overall workspace environment .

*Shikra (Opera):*

the image feature a large orange tabby cat sit on top of a wooden desk . the cat be position near the center of the desk and appear to be look at the camera . the desk be clutter with various item , include a computer , a cell phone , a mouse , a remote , and several book . some book can be spot on the left side of the desk , while others be scatter around the right side . a tv be also present in the background , mount on the wall above the desk .

*Shikra (Beam Search+TAME):*

the image feature a large orange tabby cat sit on a wooden desk , look directly at the camera . the cat be position towards the center of the desk , occupy a significant portion of the scene . the desk be clutter with various item , include a computer monitor on the left

*Shikra (OPERA+TAME):*

the image feature an orange tabby cat sit on a wooden desk , surround by a variety of object . the cat be in the center of the scene , face towards the right . there be two tv in the background ; one be position above the other and locate to the left of the cat .

*Instruction:* Please describe this image in detail

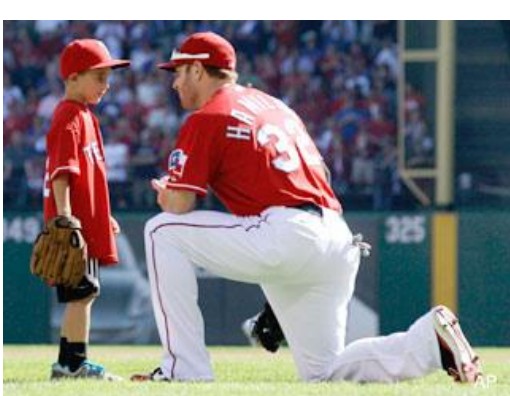

**Input Image**

*Shikra (Beam Search):*

the image capture a baseball game in progress , with a group of people wear red baseball uniform on the field . one man , possibly a coach , be crouch down next to a young boy who be wear a red hat . they be both hold baseball glove , prepare to make a play . in the background , there be a bench on the left side and a chair on the right . the bench be empty , while the chair be occupy by a spectator . the scene also include a baseball bat locate on the far right side , suggest that it be ready for a swing .

*Shikra (OPERA):*

The image shows a baseball player sitting on a bench with a baseball glove in his hand, watching a small child in a baseball uniform run up to him. The child is wearing a baseball cap and holding a baseball bat. The baseball player is wearing a baseball uniform with a hat and a baseball glove. The crowd in the background is made up of people watching the game.

*Shikra (Beam Search+TAME):*

the image feature a baseball field with a group of people , include a man and a boy , stand on the grass . the man be wear a baseball uniform and appear to be a baseball player , while the boy be also present on the field . they seem to be engage in a conversation or discuss something relate to the game . there be several other people in the background , some of whom be also wear baseball uniform . a baseball glove can be see on the ground , indicate that the player be likely prepare for a game or have just finish one .

*Shikra (OPERA+TAME):*

the image feature a young boy wear a baseball uniform , stand on a baseball field . he be wear a baseball glove , indicate his involvement in the sport . the boy be stand in front of a baseball field , possibly wait for his turn to play . in the background , there be several other people , possibly teammates or spectator , watch the game . some of them be stand closer to the foreground , while others be far in the background . the scene capture the excitement and anticipation of a baseball game .

Figure 11: Qualitative example.

