# OpenReview forum: "Intervening Anchor Token: Decoding Strategy in Alleviating Hallucinations for MLLMs"
_ICLR.cc/2025/Conference — ICLR 2025 Poster_

### Official Review · Reviewer_59hN · 2024-10-28

**Soundness:** 4
**Presentation:** 3
**Contribution:** 4
**Rating:** 8
**Confidence:** 5

**Summary:**

This paper proposes a new decoding method to address hallucination issues in multimodal large language models (MLLMs). The authors identify that excessive propagation of “anchor tokens”, influenced by polarized variance in the attention mechanism’s eigenspectrum, contributes to hallucinations by sidelining crucial visual information. To mitigate this, the paper introduces the Dynamic Token Propagation Mechanism (TAME), a plug-and-play decoding strategy that adjusts the eigenspectrum variance in the attention weights, limiting over-propagation without adding to inference time. Extensive experiments demonstrate that TAME effectively reduces hallucinations across multiple MLLMs and achieves great improvements over existing methods like Greedy Decoding and Visual Contrastive Decoding (VCD). The method is simple-yet-effective and easy to implement.

**Strengths:**

1. The paper is well-written and clear motivated. The authors begin by introducing hallucinations and related work, gradually diving deeper by focusing on the phenomena of token information flow and information aggregation, ultimately leading to the proposal of TAME.

2. This paper is built on a detailed and clear theoretical foundation. Although the proposed method appears simple, the step-by-step theoretical derivation strongly supports it.

3. The proposed method is simple-yet-effective, which obviously surpasses all baselines on CHAIR, POPE and GPT-4 assisted hallucination evaluation. Meanwhile, the method introduces no additional computation overhead, which is superior to existing contrastive decoding based methods.

**Weaknesses:**

1. Some of the color schemes in the figures, such as Figures 4 and 5, are prone to causing confusion. It is recommended to use colors with higher contrast and more distinct differentiation.

2. The paper begins its analysis with the phenomena of token information flow and token information aggregation, gradually developing a method to mitigate hallucinations. However, it does not yet discuss the underlying causes or triggers of these phenomena, only relying on empirical analysis. Nevertheless, the analytical approach used in the paper is still worth learning from.

3. The paper provides limited explanation of the proposed method. Although the method is simple (just one single formula), a smoother transition from the theoretical derivation is needed, along with a clear description of its application scenarios.

**Questions:**

1. In the qualitative results, it appears that the proposed method tends to make the model’s output shorter, which is expected, as the goal is to reduce hallucinated content. However, from another perspective, could this lead to insufficient perception of the image by the model, potentially missing some details? If so, how could we measure and balance this trade-off?

2. The example in Figure 6 is somewhat unclear, as it’s difficult for reviewers to determine which parts of the image or which specific patches correspond to the enhanced visual tokens. Could the authors provide a visualization of the relevant visual tokens mapped onto the image?

---

> ### Author Response · Authors · 2024-11-24
>
> > **Q1**: Some of the color schemes in the figures, such as Figures 4 and 5, are prone to causing confusion. It is recommended to use colors with higher contrast and more distinct differentiation.
>
> **A1**: I apologize for any confusion caused by the color schemes in the figures. We have refined the figures by adopting a color scheme with higher contrast and more distinct differentiation to enhance clarity in the final version.
>
> > **Q2**: The paper begins its analysis with the phenomena of token information flow and token information aggregation, gradually developing a method to mitigate hallucinations. However, it does not yet discuss the underlying causes or triggers of these phenomena, relying only on empirical analysis. Nevertheless, the analytical approach used in the paper is still worth learning from.
>
> **A2**: Thank you for your insightful suggestion! We posit that the primary triggers of token information flow and token information aggregation in MLLMs stem from the intrinsic properties of the attention mechanism. Specifically, in the shallow layers, the model aggregates critical information from the input sequence into a small subset of tokens, referred to as Anchor Tokens, through high attention weights. In the deeper layers, these tokens form localized information hubs, often at the expense of overlooking other contextual information. Our observation aligns with the findings reported in [1].
>
> Furthermore, we hypothesize that this phenomenon fundamentally traces back to the configuration of the QK parameters within the attention weight matrix, which plays a pivotal role in modulating attention distributions. Specifically, the eigenspectrum of the generated matrix $\mathrm{W}$ and variations in $\operatorname{tr}\left(\mathrm{W}^2\right)$ significantly influence the information propagation patterns across layers, leading to either token localization or uniformity.
>
> This analysis is elaborated in Section 3, titled **“When are tokens localized or uniform?”** Through theoretical derivations and empirical evidence, we demonstrate that token localization closely ties to the distribution of the eigenspectrum of $\mathrm{W}$. This finding not only provides a novel perspective for understanding the localized properties of the attention mechanism but also establishes a theoretical foundation for addressing hallucination issues in MLLMs.
>
> > **Q3**: The paper provides limited explanation of the proposed method. Although the method is simple (just one single formula), a smoother transition from the theoretical derivation is needed, along with a clear description of its application scenarios.
>
> **A3**: I apologize for any confusion caused by our initial description. To provide a clearer connection between the theoretical analysis and the proposed method, as well as to better explain its application scenarios, we have added a transitional paragraph in the revised version:
>
> “The previous analysis demonstrates that anchor tokens significantly impact the expressive capability and hallucination phenomena of MLLMs through the eigenspectrum properties of the attention weight matrix. Specifically, moderate propagation of anchor tokens enhances the model's expressivity, while over-propagation leads to hallucinations. These findings highlight that controlling the eigenspectrum of the query-key weight matrix ($\mathrm{W}$) can effectively regulate the propagation intensity of anchor tokens. Compared to intervening in $\mathrm{W} = \mathrm{W} _ {\mathrm{QK}} {\Sigma}$, we choose to directly adjust the eigenspectrum of $\mathrm{W} _ {\mathrm{QK}}$. This is because $\mathrm{W}$ introduces the covariance matrix ${\Sigma}$, which remains unchanged during inference, making it sufficient to adjust $\mathrm{W} _ {\mathrm{QK}}$ to flexibly control attention propagation. Hence, merely intervening the $\mathrm{W}_{\mathrm{QK}}$ helps the model avoid triggers for hallucinations.”

---

> > ### Author Response · Authors · 2024-11-24
> > **Continue'd...**
> >
> > > **Q4**: In the qualitative results, it appears that the proposed method tends to make the model’s output shorter, which is expected, as the goal is to reduce hallucinated content. However, from another perspective, could this lead to insufficient perception of the image by the model, potentially missing some details? If so, how could we measure and balance this trade-off?
> >
> > **A4**: Thank you for raising this important point. We conduct detailed evaluations of the generated output length across different decoding methods, as shown in the table below. The results indicate that while TAME effectively reduces hallucinated content, it maintains or even slightly increases the output length when integrated with decoding strategies like VCD, OPERA, or ICD. Specifically, on the COCO dataset, TAME achieves a balanced reduction in hallucinations without sacrificing detail richness, as demonstrated by the minimal variation or slight increase in the average length.
> >
> > **Table1**: Comparison of the hallucination mitigation performance across different methods in terms of output length.
> > | **Method**                  | **Length** |
> > |-----------------------------|------------|
> > | LLaVA-1.5                  | 100.6      |
> > | + VCD                      | 100.4      |
> > | + VCD w/ **TAME**          | 100.9      |
> > | + OPERA                    | 98.6       |
> > | + OPERA w/ **TAME**        | 98.4       |
> > | + ICD                      | 106.3      |
> > | + ICD w/ **TAME**          | 110.1      |
> >
> > Additionally, we evaluate the quality of the generated text in terms of richness, correctness, and user-friendliness using GPT-4 and GPT-4V-assisted evaluations, as shown in Figure 4 and Table 2 in the main paper. These assessments confirm that TAME not only mitigates hallucinations but also preserves or enhances the detailedness and correctness of the generated text on the COCO dataset. This demonstrates that our method achieves a good balance between reducing hallucinated content and maintaining sufficient detail, effectively addressing the potential trade-off. Figure 6 intuitively demonstrates the effectiveness of TAME in optimizing the attention mechanism across different tasks (e.g., Captioning and VQA) by reducing over-reliance on generated text and enhancing focus on the input visual regions, thereby significantly improving the quality of the generated results.
> >
> > > **Q5**: The example in Figure 6 is somewhat unclear, as it’s difficult for reviewers to determine which parts of the image or which specific patches correspond to the enhanced visual tokens. Could the authors provide a visualization of the relevant visual tokens mapped onto the image?
> >
> > **A5**: Thank you for your insightful suggestion. We provide a visualization in the final revision to illustrate the correspondence between different parts of the generated text and the relevant visual regions in the image. These results demonstrate that TAME, as a plug-and-play method, effectively guides the model to focus on key visual regions in the image. Furthermore, the visualization intuitively shows that TAME reduces over-reliance on language priors while significantly enhancing the model's understanding of visual content.
> >
> > #### **References**:
> > [1] Label Words are Anchors: An Information Flow Perspective for Understanding In-Context Learning

---

> > > ### Comment · Reviewer_59hN · 2024-11-24
> > >
> > > Thanks for your response, I will keep my positive score.

---

> > > > ### Author Response · Authors · 2024-11-25
> > > >
> > > > Thank you for your positive feedback and for taking the time to review our work. We greatly value your support and have submitted a revised version incorporating the suggested improvements.

---

### Official Review · Reviewer_tkGb · 2024-10-31

**Soundness:** 3
**Presentation:** 3
**Contribution:** 3
**Rating:** 8
**Confidence:** 3

**Summary:**

MLLMs encounter hallucination issues, where the generated text does not match the provided visual content. Previous methods have attempted to mitigate hallucinations by designing special decoding strategies, such as penalizing summary tokens, but they lack an analysis of the relationship between the model’s hallucinations and the summary mechanism. In this paper, a study and analysis of the causes of hallucinations in MLLMs are presented. A general decoding strategy, TAME, is proposed to reduce the excessive propagation of anchor tokens by dynamically intervening in the variance of feature spectra. Experiments demonstrate the correlation between feature spectra and hallucinations in MLLMs, as well as the effectiveness of TAME.

**Strengths:**

1. The motivations are strong, with in-depth analysis and derivation of the mechanisms behind hallucinations in MLLMs.
2. The proposed method, TAME, is a simple and effective plug-and-play decoding strategy.
3. The experiments are thorough, validating both the causes of hallucinations mentioned in the paper and the effectiveness of the proposed method.

**Weaknesses:**

1. The introduction mentions summary tokens but does not define them, leaving unclear the distinction between summary tokens and anchor tokens.
2. There are some typos in the text that need to be checked:
   1. Line 301: "Conversly" -> "Conversely"
   2. Line 333: "7,3" -> "7.3"

**Questions:**

See weakness.

---

> ### Author Response · Authors · 2024-11-23
>
> > **Q1**: The introduction mentions summary tokens but does not define them, leaving unclear the distinction between summary tokens and anchor tokens.
>
> **A1**: Thank you for your valuable feedback. We provide clear definitions for both below.
>
> **Summary Tokens** are a specific type of **text token** in LLMs that primarily serve to aggregate critical information from a sequence during generation and provide global guidance for subsequent token generation. They reflect the "Aggregation Pattern" inherent to LLMs, enabling the model to synthesize global context for generating coherent outputs. However, excessive reliance on the global information provided by summary tokens may cause the model to overlook original contextual or visual modality inputs, leading to hallucinations.
>
> **Anchor Tokens** are key tokens with high propagation probability in the attention mechanism, especially in multimodal tasks, where they emphasize information interaction between **multimodal tokens**. However, the over-propagation of anchor tokens can lead to an overemphasis on localized information, causing the generated content to disproportionately focus on specific objects or concepts while neglecting broader contextual cues. This imbalance in attention distribution can contribute to the emergence of hallucinations.
>
> ---
>
> > **Q2**: There are some typos in the text that need to be checked: Line 301: "Conversly" → "Conversely" Line 333: "7,3" → "7.3"
>
> **A2**: We apologize for the confusion caused by the typo. We have carefully reviewed the text and made the necessary corrections in the revised manuscript.
>
> We hope that our response sufficiently addresses your comments. If you have any further questions or concerns, please feel free to reach out to us for further discussion.

---

### Official Review · Reviewer_kinG · 2024-11-02

**Soundness:** 4
**Presentation:** 3
**Contribution:** 3
**Rating:** 6
**Confidence:** 3

**Summary:**

This paper investigates the causes of hallucinations in LVLMs through the attention weight matrix of anchor tokens. Then this paper demonstrates the propagation pattern of anchor tokens by the eigenspectrum of the attention weight matrix. The authors further propose a versatile plug-and-play decoding strategy named Dynamic Token Propagation Mechanism (TAME) to reduce the over-propagation of anchor tokens through dynamically intervening in the eigenspectrum variance. Extensive experiments show that TAME improves hallucination metrics across multiple MLLMs.

**Strengths:**

1.	This paper provides a solid theoretical framework by analyzing the relationship between LVLM hallucinations and attention patterns.
2.	This paper introduces a novel plug-and-play decoding strategy that dynamically adjusts the eigenspectrum variance of attention weights to mitigate hallucinations without adding inference time.
3.	Extensive experiments show that TAME improves hallucination metrics across multiple MLLMs.
4.	TAME can be integrated into various decoding strategies without requiring additional training or data.

**Weaknesses:**

1.	Although this paper provides an additional theoretical framework, the impact of anchor tokens on the LVLM hallucination was first proposed by OPERA [1].
2.	TAME primarily addresses object hallucinations but may not be as effective for other hallucination types, such as incorrect attributes or relations (e.g., limited performance improvement on the MME benchmark). Experiments on more types of hallucination should be conducted.
3.	It would be interesting to study applying TAME to some layers rather than all layers through experiments.

[1] OPERA: Alleviating Hallucination in Multi-Modal Large Language Models  via Over-Trust Penalty and Retrospection-Allocation

**Questions:**

Please see the weakness.

---

> ### Author Response · Authors · 2024-11-23
>
> > **Q1**: Although this paper provides an additional theoretical framework, the impact of anchor tokens on LVLM hallucination was first proposed by OPERA [1].
>
> **A1**: Thank you for your suggestion. OPERA [1] is a relatively complex approach to reducing hallucinations, focusing primarily on anchor tokens in the textual modality without fully considering the deeper implications of multimodal interactions in MLLMs. While OPERA effectively reduces hallucinations through a penalty mechanism targeting anchor token contributions, it incurs significant computational overhead, increasing inference time by 2-3 times due to the additional processing required for each token.
>
> In contrast, our proposed method, TAME, offers an efficient solution that outperforms current state-of-the-art approaches. TAME not only accounts for the interactions between multimodal tokens but also addresses the limitations in existing methods regarding the understanding of anchor tokens' roles in multimodal tasks. Specifically, the dual nature of anchor tokens, which can act as both a "blessing" and a "curse," has not been sufficiently clarified. Our work focuses on two key research questions: (Q1) Under what conditions do tokens exhibit localized or uniform distributions? (Q2) How do anchor tokens influence the generation of hallucinations?
>
> Moreover, TAME mitigates the over-propagation of anchor tokens by dynamically adjusting the eigenspectrum variance of the attention weight matrix. This lightweight approach significantly improves hallucination reduction without introducing additional inference costs. Our work provides a novel and theoretically grounded framework for addressing hallucination problems in MLLMs, demonstrating its efficiency and robustness across practical tasks.
>
> > **Q2**: TAME primarily addresses object hallucinations but may not be as effective for other types of hallucinations, such as incorrect attributes or relations (e.g., limited performance improvement on the MME benchmark). Experiments on more types of hallucination should be conducted.
>
> **A2**: Thank you for your insightful comments. The table below presents our experimental results on the hallucination subset of the MME dataset, which includes object-level hallucinations and attribute-level hallucinations. TAME, as a plug-and-play decoding strategy, delivers significant performance improvements across methods by dynamically adjusting the eigenspectrum variance of the attention weight matrix. This mitigates anchor token over-propagation and enhances model performance in object existence, count estimation, position alignment, and color consistency.
>
> **Table 1**: Evaluation results on the hallucination subset of MME. Object-level: *Existence* and *Count*. Attribute-level: *Position* and *Color*. max-tokens=512.
> | **Decoding**      | *Existence* ↑ | *Count* ↑ | *Position* ↑ | *Color* ↑ | **Total Scores** ↑ |
> |--------------------|---------------|------------|--------------|------------|--------------------|
> | **LLaVA-1.5-7B**   | 163.67        | 114.66     | 104.00       | 141.00     | 523.33            |
> | + Greedy           | 184.00        | 95.33      | 112.00       | 157.67     | 549.00            |
> | + VCD              | 172.67        | 120.33     | 129.67       | 155.00     | 561.33            |
> | + OPERA            | 174.67        | 115.66     | 110.67       | 141.00     | 542.00            |
> | + OPERA w/ **TAME**| 176.00        | 118.33     | 113.00       | 143.00     | 550.33            |
> | + SID              | 182.00        | 127.00     | 116.00       | 139.00     | 564.00            |
> | + SID w/ **TAME**  | **193.00**    | **137.33** | **139.00**   | **164.67** | **634.00**        |
>
> ---

---

> ### Author Response · Authors · 2024-11-23
> **Continue'd...**
>
> > **Q3**: It would be interesting to study the application of TAME to some layers rather than all layers through experiments.
>
> **A3**: Thank you for your insightful suggestion. As analyzed in Figure 1 and Appendix A (main paper), early, middle, and late layers exhibit significant differences in token propagation patterns. Early layers (**$l_{0}$**) primarily focus on extracting low-level features and token initialization, middle layers (**$l_{16}$**) emphasize multi-modal alignment and feature aggregation, while late layers (**$l_{31}$**) are responsible for high-level reasoning and final output generation. These differences play a critical role in the model's behavior and error generation, especially in hallucination-prone scenarios.
>
> We evaluate TAME by applying it to different combinations of layers (**$l_{0}$**, **$l_{16}$**, and **$l_{31}$**), as shown in the table below. In this experiment, the baseline is OPERA [1] (**Exp I**). The results from Exp II-IV demonstrate the incremental impact of TAME when applied to individual layers: **Early layers (**$l_{0}$**, Exp II)**: Applying TAME at this stage slightly reduces hallucinations by refining token initialization and propagation, but its overall impact is limited due to the lack of deeper-layer information.**Middle layers (**$l_{16}$**, Exp III)**: Incorporating TAME at this stage significantly improves multi-modal alignment, optimizing feature integration and enhancing response generation accuracy.**Late layers (**$l_{31}$**, Exp IV)**: Applying TAME here substantially reduces errors, as this stage directly influences final reasoning and output generation. The results from Exp V-VIII further demonstrate the cumulative effect of applying TAME to multiple layers: **$l_{0} + l_{16} + l_{31}$ (Exp VIII)**: Applying TAME across all layers achieves the best overall performance, with the fewest hallucinations and the highest accuracy across all tasks.
>
> This phenomenon highlights the importance of holistically optimizing token propagation across the model. Early layers provide foundational improvements, middle layers optimize multi-modal representations, and late layers ensure high-level reasoning accuracy. The experiments suggest that integrating TAME across more layers significantly reduces error-prone responses, as it dynamically mitigates over-propagation at different stages of the model.
>
> **Table2**: Ablation study of applying TAME on different layers. **$l_{0}$**: layer 0; **$l_{16}$**: layer 16; **$l_{31}$**: layer 31
> | **Exp** | **$l_{0}$** | **$l_{16}$** | **$l_{31}$** | **MMB** | **GQA** | **CHAIR$_{S}$** | **CHAIR$_{I}$** |
> |:-------:|:-----------:|:------------:|:------------:|:-------:|:-------:|:--------------:|:--------------:|
> | I       |             |              |              | 64.8    | 62.0    | 46.4           | 13.0           |
> | II      | **✔**       |              |              | 64.9    | 62.0    | 46.2           | 12.7           |
> | III     |             | **✔**        |              | 64.8    | 62.1    | 45.8           | 12.6           |
> | IV      |             |              | **✔**        | 64.9    | 62.3    | 45.9           | 12.7           |
> | V       | **✔**       | **✔**        |              | 65.1    | 62.3    | 44.2           | 12.6           |
> | VI      | **✔**       |              | **✔**        | **65.3**| **62.6**| 44.4           | 12.5           |
> | VII     |             | **✔**        | **✔**        | 65.2  | 62.4    | 43.9         | 12.3         |
> | VIII    | **✔**       | **✔**        | **✔**        | **65.3**| 62.5  | **41.3**       | **12.2**       |
>
>
> #### **References**:
> [1] *OPERA: Alleviating hallucination in multi-modal large language models via over-trust penalty and retrospection-allocation.*

---

> > ### Comment · Reviewer_kinG · 2024-11-24
> >
> > Thanks for your response, I will keep my positive score.

---

> > > ### Author Response · Authors · 2024-11-24
> > >
> > > Thank you for your positive feedback and support! We appreciate your insights and have submitted a revised version addressing the comments provided.

---

### Official Review · Reviewer_UP4R · 2024-11-07

**Soundness:** 3
**Presentation:** 2
**Contribution:** 3
**Rating:** 6
**Confidence:** 3

**Summary:**

This paper explores the role of anchor tokens in causing hallucinations in LLMs, and proposed Dynamic Token Propagation Mechanism (TAME) based on eigenspectrum variance of the attention weight. The results show that TAM can be integrated with existing decoding method including Beam/VCD/ICD/SID and improve the performance consistently on InstructBLIP, MiniGPT-4, LLaVA-1.5, and Shikra.

**Strengths:**

1. Simple method which is very easy to use
2. Good insights from eigenspectrum variance
3. Constantly improve the decoding methods of InstructBLIP, MiniGPT-4, LLaVA-1.5, and Shikra.

**Weaknesses:**

1. I hope the paper could explore additional approaches to reduce hallucinations, such as Retrieval-Augmented Generation (RAG), Reinforcement Learning from Human Feedback (RLHF), improvements in factuality metrics, and high-quality, data-based instruction tuning.

2. The paper currently uses the a few simple eval sets like MS COCO. I would suggest incorporating more challenging benchmarks for a more rigorous assessment.

3. The writing could benefit from refinement. For instance, many citations are not in the correct format.

4. The paper only reports metrics for hallucination. But it is not clear the new predictions do better or worse in other metrics. Sometimes less hallucination may lead to less details or not as friendly to users to read.

**Questions:**

1. in Table 1: the baselines with max token 512 generate very high hallucination (>30%). Is it aligned with user's experience? At least I feel Gemini and ChatGPT's numbers should be much better than that.

2. I feel the proposed method can be used for any LLM based method. Why does the title claim it only for multimodal LLMs?

---

> ### Author Response · Authors · 2024-11-23
>
> > **Q1**: I hope the paper could explore additional approaches to reduce hallucinations, such as Retrieval-Augmented Generation (RAG), Reinforcement Learning from Human Feedback (RLHF), improvements in factuality metrics, and high-quality, data-based instruction tuning.
>
> > **Q2**: The paper currently uses a few simple evaluation sets like MS COCO. I would suggest incorporating more challenging benchmarks for a more rigorous assessment.
>
> **A1 and A2**: Thank you for your insightful suggestion. We conduct a rigorous evaluation of VCD, OPERA, and RAG (Vanilla-RAG [1], SURf-RAG [2]) as well as RLHF (RLHF-V [3], CSR [4]) on six popular MLLM benchmarks and four additional ones, as shown in the table below.
>
> For details, **SEED-Bench** consists of 19k multiple choice questions with human annotations, while spanning 12 evaluation dimensions. **GQA** incorporates a novel evaluation metrics suite focused on consistency, grounding, and plausibility, establishing a rigorous standard for assessing in vision-language tasks. **Vizwiz** examines certain perception capabilities, like knowledge and relation. **MME** contains 14 meticulously designed subtasks that challenge the models' interpretative and analytical skills. **MMBench** examines LVLMs on general perception capabilities using a wide range of tasks. **POPE** is an assessment methodology designed to scrutinize object hallucination in LVLMs.
>
> **Vanilla-RAG** concatenates the Top-N image-caption pairs from the database, which have the highest CLIP score similarity to the test image, before appending the questions and images for the LVLMs to respond. **SURf-RAG** is a self-refinement framework that teaches LVLMs to selectively utilize retrieved information. **RLHF-V** collects fine-grained paragraph-level corrections from humans on hallucinations and performs dense direct preference optimization using human feedback. **CSR** enables the model to self-improve by iteratively generating candidate responses, evaluating the reward for each response, and curating preference data for fine-tuning.
>
> For the RAG and RLHF methods, our approach seamlessly integrates as a plug-and-play module within their frameworks, without incurring extra inference time. By combining RAG with TAME, external knowledge retrieval enhances the accuracy of generated outputs, ensuring better alignment with the input context. Similarly, when integrated with RLHF, our method leverages human feedback to guide the model in producing outputs that are more faithful to the actual content. Experimental results demonstrate that our approach delivers comprehensive improvements across both frameworks, further validating its effectiveness.
>
> We conduct additional experiments to evaluate improvements in factuality metrics and the effectiveness of data-driven instruction tuning on model performance. Our evaluations focus on benchmarks such as **GQA**, **Vizwiz**, **MME**, and **POPE**, which test real-world knowledge QA and multimodal understanding tasks, as shown in the table below. The results demonstrate that instruction tuning with LRV-Instruction-finetuned [5] moderately enhances performance by leveraging high-quality image-text pairs for task-specific fine-tuning, improving the model's alignment with real-world knowledge. However, integrating TAME further amplifies these gains by dynamically mitigating hallucinations and strengthening factual alignment, resulting in significant improvements across all benchmarks. This combination achieves a higher degree of correctness and consistency in generated outputs, validating that TAME effectively complements instruction tuning as a plug-and-play enhancement, improving both accuracy and robustness in real-world multimodal tasks.

---

> > ### Comment · Reviewer_UP4R · 2024-12-01
> >
> > Thanks for the detailed reply. I would like to thank authors for the consideration and upgrade my rating to 6 marginally above the acceptance threshold

---

> > > ### Author Response · Authors · 2024-12-01
> > >
> > > Thank you for your valuable feedback and constructive suggestions. We truly appreciate the time and effort you have dedicated to reviewing our work. Based on your insights, we have carefully revised our manuscript and addressed all the comments provided.

---

> ### Author Response · Authors · 2024-11-23
> **Continue'd...**
>
> **Table 1**: Comparison of methods on different benchmarks. **SEEDᴵ** refers to SEED *image* evaluation, **VisWizⱽ** refers to *image* evaluation for VisWiz VQA, **POPEᴿ** refers to POPE Random, **MMEᴼ** refers to MME Object-level Hallucination Existence Count.
>
> | **Methods**                     | **GQA** | **SEEDᴵ** | **VisWizⱽ** | **MMEᴼ** | **MMEᴬ** | **MMB** | **POPEᴿ** |
> |----------------------------------|---------|-----------|-------------|----------|----------|---------|-----------|
> | **LLaVA-1.5-7B**                | 60.4    | 58.1      | 49.0        | 278.33   | 245.00   | 64.2    | 83.6      |
> | + VCD                           | 61.0    | 58.9      | 50.8        | 293.00   | 268.33   | 61.4    | 84.9      |
> | + VCD w/ **TAME**               | 61.7    | 59.4      | 51.6        | **295.67**| **275.67**| 61.5    | 86.0      |
> | + OPERA                         | 62.0    | 59.6      | 52.4        | 290.33   | 251.67   | 64.8    | 85.4      |
> | + OPERA w/ **TAME**             | 62.5    | **60.7**  | 52.9        | 294.33 | 256.00   | *65.3*  | 89.0      |
> | + Vanilla-RAG                   | 60.2    | -         | 49.6        | 264.33   | 255.33   | -       | 85.7      |
> | + Vanilla-RAG w/ **TAME**       | 61.8    | -         | 50.7        | 272.33   | 259.67   | -       | 88.5      |
> | + SURf-RAG                      | 62.4    | -         | 54.3      | 268.67   | 253.00   | -       | 87.9      |
> | + SURf-RAG w/ **TAME**          | **62.9**| -         | **55.2**    | 270.00   | 258.33   | -       | 89.2    |
> | + RLHF-V                        | 62.3    | 59.3      | 53.7        | 283.33   | 263.67   | 63.6    | 86.2      |
> | + RLHF-V w/ **TAME**            | 62.8 | 59.8      | 54.2        | 272.33   | 266.33   | 64.2    | 87.1      |
> | + CSR                           | 61.9    | 60.0      | 53.4        | 285.67   | 264.67   | 65.2    | 87.0      |
> | + CSR w/ **TAME**               | 62.4    | 60.6    | 53.9        | 294.00   | 271.00 | **66.9**| **89.5**  |
> | **LLaVA-1.5-13B**               | 64.1 | -         | 53.3        | -        | -        | 68.2    | 86.5      |
> | + OPERA                         | 64.0    | -         | 55.6      | -        | -        | **68.9**| 87.2    |
> | + OPERA w/ **TAME**             | **65.7**| -         | **56.2**    | -        | -        | 68.8  | **89.0**  |
> | **LLaVA-NeXT-Mistral-7B**       | 62.9    | -         | 52.6        | -        | -        | 66.1    | 88.2      |
> | + OPERA                         | 63.4  | -         | *52.9*      | -        | -        | 67.2  | 88.7    |
> | + OPERA w/ **TAME**             | **64.6**| -         | **54.0**    | -        | -        | **67.4**| **90.4**  |
> | **mPLUG-Owl-7B**                     | 66.7 | 63.5 | 57.1   | 310.33  | 281.67   | 68.2    | 89.1  |
> | + LRV-Instruction-finetuned          | 67.5 | 64.2 | 58.0   | 315.00  | 285.33   | 69.5    | 90.0  |
> | + LRV-Instruction-finetuned w/ **TAME**  | **68.9** | **65.8** | **59.2**   | **322.33**  | **291.00**   | **71.0**    | **91.5**  |
>
> ---

---

> > ### Author Response · Authors · 2024-11-23
> > **Continue'd...**
> >
> > > **Q3**: The writing could benefit from refinement. For instance, many citations are not in the correct format.
> >
> > **A3**: Thank you for your valuable feedback. We carefully review the manuscript and refine the writing. Specifically, we address the issues with citation formatting and ensure that all references are presented correctly.
> > > **Q4**: The paper only reports metrics for hallucination. However, it is not clear whether the new predictions perform better or worse in other metrics. Sometimes, reducing hallucination may result in less detail or make the output less user-friendly to read.
> >
> > **A4**: Thank you very much for your thoughtful suggestions. As shown in Figure 4 and Table 2, our experiments evaluate not only hallucination-related metrics but also the overall quality of the generated text through GPT-4 and GPT-4V assisted evaluations, focusing on detail richness and correctness.
> >
> > Specifically, in the GPT-4 assisted evaluation, we analyze six key metrics using the VG-100K dataset, as illustrated in Figure 4 in the paper. These include **Sentences per Image (SPI)** and **Words per Image (WPI)**, which measure the richness and detail of the text; **Hallucinated Sentences per Image (HSPI)** and **Hallucinated Words per Image (HWPI)**, which quantify the frequency of hallucinated sentences and words; and **Hallucinated Sentences Ratio (HSR)** and **Hallucinated Words Ratio (HWR)**, which evaluate the proportion of hallucinated content in the generated text. Through the GPT-4V assisted evaluation, we further assess correctness using the MSCOCO dataset. The results based on these metrics are as follows: Our method significantly reduces hallucinations, as evidenced by notable decreases in HSPI and HSR, while maintaining a high level of text detail, with minimal changes in SPI and WPI. This demonstrates that our approach effectively reduces hallucinations without compromising detail richness.
> >
> > As shown in Table 2 in the paper, the GPT-4V evaluation includes two dimensions: **Correctness**, which measures whether the generated text accurately reflects the content of the image, and **Detailedness**, which evaluates whether the descriptions contain sufficient detail. The results demonstrate that TAME reduces hallucinations while maintaining or even enhancing the level of detail and user-friendliness of the generated text. These evaluations collectively indicate that TAME not only reduces hallucinations but also generates detailed text.
> >
> > In Appendix G.6, we comprehensively assess the overall quality of the generated text. Using Perplexity (PPL), a classical metric in NLP that does not rely on reference text, we evaluate the grammar, fluency, and naturalness of the generated text with the assistance of GPT-4. To ensure robustness, we randomly select 1,000 images from the MSCOCO dataset and conduct evaluations on the LLaVA-1.5 7B model. The results demonstrate that TAME consistently improves the quality of the generated text across various aspects.

---

> > > ### Author Response · Authors · 2024-11-23
> > >
> > > > **Q5**: In Table 1, the baselines with a max token limit of 512 generate very high hallucination (>30%). Is this aligned with users' experience? At least, I feel that Gemini and ChatGPT's numbers should be much better than that.
> > >
> > > **A5**: Thank you for your insightful question. Our experimental results derive from rigorous evaluation methods and challenging test datasets, such as MSCOCO and VG-100K. These datasets encompass complex multimodal scenarios that require models to possess a high level of understanding and generation capability. The observed high hallucination rates primarily reflect the limitations of existing baseline models in handling complex tasks. However, evaluations conducted using GPT-4 and GPT-4V show that, on the current open-source models, our TAME approach not only reduces hallucinations but also generates detailed text.
> > >
> > > Moreover, the significant performance gap between open-source models and closed-source models (such as ChatGPT and Gemini) arises from several factors. Closed-source models typically leverage much larger training datasets, enabling them to cover a broader range of scenarios. Additionally, these models often employ more advanced architectures, enhancing their generalization ability and significantly reducing the likelihood of generating hallucinations.
> > >
> > > To further investigate this phenomenon, we conduct experiments on models of different scales, including **LLaVA-1.5-7B**, **LLaVA-1.5-13B**, and **LLaVA-NeXT-Mistral-7B**. The experimental results presented in the table 1 show that smaller models (such as **LLaVA-1.5-7B**) exhibit significantly higher hallucination rates, while larger models (such as **LLaVA-1.5-13B**) demonstrate much lower rates. Based on these findings, we draw the following conclusions:
> > >
> > > 1. **Relationship Between Model Scale and Hallucinations**: Due to their limited parameter capacity, smaller models (e.g., **LLaVA-1.5-7B**) struggle to integrate and generalize multimodal information effectively. This limitation causes them to rely more heavily on internal priors, leading to higher hallucination rates. In contrast, larger models (e.g., **LLaVA-1.5-13B**) benefit from richer parameterization and stronger representation capabilities, allowing them to align visual and textual modalities more effectively and reduce dependency on erroneous or fabricated information.
> > >
> > > 2. **Data Utilization and Generalization**: Open-source models are often constrained by the limited scale of their training data. Compared to closed-source models, they exhibit weaker generalization capabilities in complex or ambiguous scenarios, which further exacerbates hallucination generation.
> > >
> > > 3. **Comparison with Closed-Source Models**: Closed-source models such as **ChatGPT** and **Gemini** utilize proprietary datasets and advanced techniques (e.g., **Mixture-of-Experts**) to dynamically allocate resources for specific tasks. This approach ensures a more contextually aware generation process and significantly reduces hallucination rates, especially in complex tasks.

---

> ### Author Response · Authors · 2024-11-23
> **Continue'd...**
>
> > **Q6**: I feel the proposed method can be used for any LLM-based approach. Why does the title claim it is only for multimodal LLMs?
>
> **A6**: Thank you for your question. We extend our experiments to single-modal LLMs. As shown in the table below, we conduct evaluations on the **Wikitext-103** [6] and **MiniPile** [7] datasets to assess the scalability and consistency of TAME's impact. TAME was integrated as a plug-and-play enhancement across three distinct model configurations, including **BLOOM** (LLaMA architecture with ALiBi) [8] and **OpenLLaMA** [9].
>
> The results show that TAME consistently improves perplexity (PPL) across all architectures and parameter sizes, demonstrating its effectiveness in enhancing model performance. These findings further support that TAME is a generalizable mechanism, suitable for both multimodal and single-modal LLMs, broadening its applicability.
>
> **Table 2**:Perplexity (PPL) comparison on WikiText-103 and MiniPile datasets using BLOOM and OpenLLaMA architectures with and without TAME across varying parameter sizes.
> | **WikiText-103**               |        |       | **MiniPile**               |        |       |
> |--------------------------------|--------|-------|----------------------------|--------|-------|
> | **Model**                      | **#Params** | **PPL** | **Model**                | **#Params** | **PPL** |
> | BLOOM                          | 71M    | 29.9  | BLOOM                     | 160M   | 25.8  |
> | BLOOM w/ **TAME**              | 71M    | **29.0** | BLOOM w/ **TAME**        | 160M   | **25.3** |
> | OpenLLaMA                      | 71M    | 27.4  | OpenLLaMA                 | 160M   | 25.9  |
> | OpenLLaMA w/ **TAME**          | 71M    | **26.9** | OpenLLaMA w/ **TAME**    | 160M   | **24.9** |
> | BLOOM                          | 160M   | 27.6  | BLOOM                     | 430M   | 20.6  |
> | BLOOM w/ **TAME**              | 160M   | **26.0** | BLOOM w/ **TAME**        | 430M   | **19.3** |
> | OpenLLaMA                      | 160M   | 22.5  | OpenLLaMA                 | 430M   | 19.6  |
> | OpenLLaMA w/ **TAME**          | 160M   | **21.3** | OpenLLaMA w/ **TAME**    | 430M   | **19.4** |
>
> ---
>
> #### **References**:
> [1] *Dense passage retrieval for open-domain question answering.*
>
> [2] *SURf: Teaching Large Vision-Language Models to Selectively Utilize Retrieved Information.*
>
> [3] *RLHF-V: Towards trustworthy MLLMs via behavior alignment from fine-grained correctional human feedback.*
>
> [4] *Calibrated Self-Rewarding Vision Language Models.*
>
> [5] *Mitigating Hallucination in Large Multi-Modal Models via Robust Instruction Tuning.*
>
> [6] *Pointer sentinel mixture models.*
>
> [7] *The minipile challenge for data-efficient language models.*
>
> [8] *The power of scale for parameter-efficient prompt tuning.*
>
> [9] *LLaMA: Open and efficient foundation language models.*

---

> > ### Author Response · Authors · 2024-11-26
> >
> > Dear Reviewer UP4R,
> >
> > We sincerely appreciate the time and effort you have invested in reviewing our submission. Your insightful feedback has been invaluable to us, and we have diligently worked to address all the concerns you raised in our rebuttal. As the author-reviewer discussion phase is drawing to a close, we would like to confirm whether our responses have effectively addressed your concerns. We are more than happy to provide any further details or explanations. Thank you once again for your thoughtful review and consideration.
> >
> > Best regards,
> >
> > The Authors

---

### Meta-Review · Area_Chair_Tbdm · 2024-12-11

**Metareview:**

The paper proposes a mitigation strategy for hallucinations in MLLMs, by analyzing and counteracting over-localization of attention patterns over so-called anchor tokens, and their eigenvalue scale. They show the effectiveness of their method in multiple settings and metrics. Requests for further experiments and ablations are addressed well, as noted by the reviewers.

**Additional Comments On Reviewer Discussion:**

Reviewers and authors engaged in discussion and concerns were addressed well

---

### Decision · Program_Chairs · 2025-01-22

Accept (Poster)